# A sulfur-aromatic gate latch is essential for opening of the Orai1 channel pore

Priscilla S-W Yeung[1†], Christopher E Ing[2,3†], Megumi Yamashita[1], Régis Pomès[2,3], Murali Prakriya[1]*

[1]Department of Pharmacology, Northwestern University, Feinberg School of Medicine, Chicago, United States; [2]Molecular Medicine, Hospital for Sick Children, Toronto, Canada; [3]Department of Biochemistry, University of Toronto, Toronto, Canada

**Abstract** Sulfur-aromatic interactions occur in the majority of protein structures, yet little is known about their functional roles in ion channels. Here, we describe a novel molecular motif, the M101 gate latch, which is essential for gating of human Orai1 channels via its sulfur-aromatic interactions with the F99 hydrophobic gate. Molecular dynamics simulations of different Orai variants reveal that the gate latch is mostly engaged in open but not closed channels. In experimental studies, we use metal-ion bridges to show that promoting an M101-F99 bond directly activates Orai1, whereas disrupting this interaction triggers channel closure. Mutational analysis demonstrates that the methionine residue at this position has a unique combination of length, flexibility, and chemistry to act as an effective latch for the phenylalanine gate. Because sulfur-aromatic interactions provide additional stabilization compared to purely hydrophobic interactions, we infer that the six M101-F99 pairs in the hexameric channel provide a substantial energetic contribution to Orai1 activation.

*For correspondence:
m-prakriya@northwestern.edu

†These authors contributed equally to this work

Competing interests: The authors declare that no competing interests exist.

## Introduction

The opening and closing of ion channels constitute an important means by which extracellular signals are translated into the activation of specific intracellular signaling cascades. Critical to this process is tight control of the channel gate, which forms an ion-proof barrier at rest and permits ion conduction when activated. In this study, we describe an essential molecular motif involving a sulfur-aromatic interaction that plays a critical role in the gating of $Ca^{2+}$ release-activated $Ca^{2+}$ (CRAC) channels formed by Orai1. CRAC channels are activated by the engagement of cell surface receptors that activate phospholipase C through G-proteins or tyrosine kinase cascades to cleave phosphatidylino-sitol 4,5-bisphosphate ($PIP_2$) and produce soluble inositol 1,4,5-trisphosphate ($IP_3$). The ensuing depletion of endoplasmic reticulum (ER) $Ca^{2+}$ stores activates the ER $Ca^{2+}$ sensors, STIM1 and STIM2, which translocate to the junctional ER to interact with and activate CRAC channels encoded by the Orai1-3 proteins. During this process, STIM1, which is thought to be compactly folded at rest, adopts an extended multimeric conformation to expose the Orai1-activating domain (CAD), making it available for binding to Orai1 channels at ER-plasma membrane junctions (*Prakriya and Lewis, 2015*).

Key hallmarks of CRAC channels include store-dependent activation, exquisite $Ca^{2+}$ selectivity, and a low unitary conductance, making them ideally suited for generating oscillatory $Ca^{2+}$ signals and long-lasting $[Ca^{2+}]_i$ elevations needed for transcriptional and enzymatic cascades (*Prakriya and Lewis, 2015*). In human patients with mutations in the genes encoding the prototypic CRAC channel proteins, Orai1 or STIM1, defects in $Ca^{2+}$ signaling lead to severe combined immunodeficiency, autoimmunity, ectodermal dysplasia, and tubular aggregate myopathy (*Lacruz and Feske, 2015*). The prominent role of CRAC channels for human immunity and host-defense mechanisms has led to

their emergence as drug targets for inflammatory diseases (*Stauderman, 2018*) and spurred strong interest in elucidating the molecular underpinnings of the gating mechanism (*Yeung et al., 2020*).

Crystal structures of the highly homologous *Drosophila melanogaster* Orai (dOrai) and recent concatemer-based studies have shown that Orai channels are formed by six subunits (*Cai et al., 2016*; *Hou et al., 2012*; *Liu et al., 2019*; *Yen et al., 2016*). Each of these protomers has four trans-membrane domains (TMs) which are together arranged in three concentric layers, with TMs 2–4 surrounding a narrow pore lined by six TM1 helices (*Figure 1*; *Hou et al., 2012*). Previous studies support a model wherein STIM1 binding to the cytosolic surface of Orai1 initiates a conformational wave throughout the protein (*Yeung et al., 2020*; *Zhou et al., 2019*), ultimately culminating in rotation of the pore helices to activate a hydrophobic gate in the outer pore (*Yamashita et al., 2017*). Together with a modest dilation of the outer pore (*Yeung et al., 2018*), the displacement of F99 residues away from the pore axis lowers the overall energetic penalty in this region to permit pore hydration and ion conduction (*Figure 1A*; *Yamashita et al., 2017*). Although the hydrophobic gate also involves the neighboring residue V102 located one turn above (*McNally et al., 2012*; *Yeung et al., 2017*), and may even extend to L95 located below, for simplicity, we will refer to the hydrophobic gate as the 'F99 gate' because F99 is the residue whose movement has been shown to directly contribute to pore opening (*Yamashita et al., 2017*).

The nature of the proximal signal that facilitates displacement of the F99 gate away from the pore axis during channel activation is not known. However, one crucial relay along the gating pathway to the pore is a cluster of tightly packed hydrophobic residues at the interface between the pore helices and its surrounding TM2/3 ring (*Yeung et al., 2018*). In the crystal structure of dOrai,

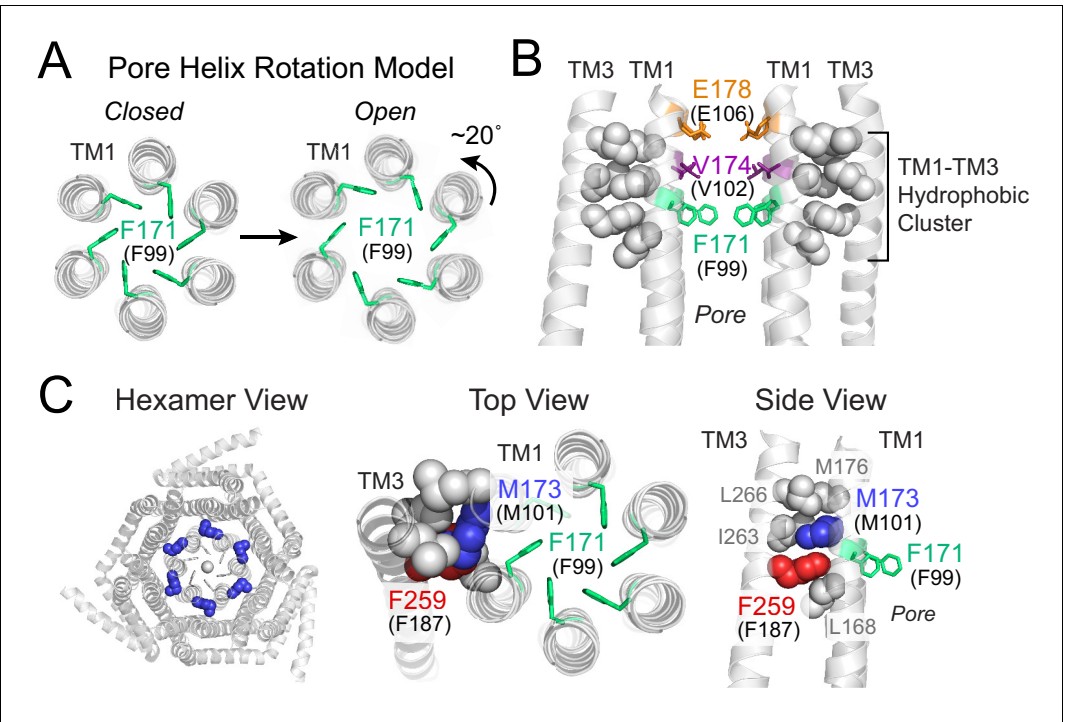

**Figure 1.** Gating model of Orai1 involving reorientation of the F99 gate. (**A**) Schematic cartoon of the pore helix rotation model. Gating occurs through a modest rotation of TM1 and dilation of the outer pore (*Yeung et al., 2018*), which moves F171 (hOrai1 F99) away from the pore axis and decreases the free energy barrier in the hydrophobic stretch for ion permeation. For simplicity, TMs 2–4 are not shown. (**B**) A cluster of hydrophobic residues (shown as gray spheres) between the TM1 pore helices and the TM3 segment. The selectivity filter E178 (hOrai1 E106, orange) and hydrophobic gate residues V174 (hOrai1 V102, purple) and F171 (hOrai1 F99, green) in the pore are also shown. Four TM1 helices and two TM3 helices are shown for clarity. (**C**) Position of M101 within the hydrophobic cluster. F171 (hOrai1 F99) is also shown within the context of the hexameric channel and relative to the hydrophobic cluster residues. Residues M173 (hOrai1 M101) and F259 (hOrai1 F187) are depicted as blue and red spheres, respectively. The F171 gate is shown as green sticks. hOrai1 numbering is shown in parentheses.

this hydrophobic cluster at the TM1-TM3 interface is formed by the TM1 residues with dOrai residue numbering L168, M173, M176 and TM3 residues F259, I263, L266 (equivalent to human Orai1 L96, M101, M104, F187, V191, L194) (*Figure 1B,C*). We previously showed that mutations that lower the hydrophobicity or reduce the side-chain size of these residues lead to loss-of-function channel phenotypes, suggesting that the interwoven hydrophobic interactions in this region function as a rigid relay during allosteric gating by STIM1 (*Yeung et al., 2018*). In the current study, we identify an additional novel function for one of these residues, M101 (*Figure 1C*), whose robust sulfur-aromatic interaction with the F99 channel gate actively stabilizes the pore in its open configuration.

## Results

### Molecular dynamics simulations reveal sulfur-aromatic interactions involving the hydrophobic gate in activated states

To better understand the molecular mechanisms underlying F99 gate opening, we began our studies by carefully mapping the movements of F99 in molecular dynamics (MD) simulations of gain-of-function (GOF) and loss-of-function (LOF) dOrai mutants (*Figure 2*). Although there are now several available structures of activating mutants (*Hou et al., 2018*; *Liu et al., 2019*), we performed simulations using the 3.35 Å crystal structure of *Drosophila melanogaster* Orai ([PDB ID:4HKR]; *Hou et al., 2012*) because it remains the most complete, highest-resolution structure and has been utilized in previous MD studies (*Bonhenry et al., 2019*; *Dong et al., 2019*). Further, given the high sequence identity between dOrai and human Orai1 (hOrai1) within the transmembrane domains, channel phenotypes identified using MD simulations in dOrai are likely to be translatable to electrophysiological results in hOrai1.

For our simulations, we took advantage of mutations at a well-studied gating locus, H134 on TM2 (equivalent to dOrai H206), to model inactive, partially active, or fully active channels. The histidine side-chain of H134 faces the non-pore-lining surface of TM1 and has been suggested to stabilize the closed channel state by acting as a steric brake at the TM1-TM2/3 ring interface (*Yeung et al., 2018*; *Figure 2—figure supplement 1A*). Previous studies have described several interesting GOF or LOF mutations at hOrai1 H134 that either inhibit or constitutively activate Orai1 gating (*Frischauf et al., 2017*; *Yeung et al., 2018*). Compared to WT channels, the LOF mutant hOrai1 H134Y (dOrai H206Y) conducts small, non-selective Orai1 currents when co-expressed with STIM1, reflecting a defect in STIM1-mediated gating (*Yeung et al., 2018*). A second mutant, hOrai1 H134Q (dOrai H206Q), is modestly active at rest but can be further gated by STIM1. By contrast, a third mutant, hOrai1 H134C (dOrai H206C), is one of the most strongly active variants, displaying similar properties as STIM1-gated channels even without STIM1, including inward-rectifying, $Ca^{2+}$-selective currents and displacement of the F99 gate (*Bulla et al., 2019*; *Frischauf et al., 2017*; *Yeung et al., 2018*). The channel activities of these mutants, as measured by electrophysiology (*Figure 2—figure supplement 1B*), correlate well with metrics such as the extent of pore helix rotation, outer pore dilation, and pore hydration observed in MD simulations (*Figure 2—figure supplement 2B*; [see also *Yeung et al., 2018*]).

Within these dOrai H206 simulations, we looked for conformational changes in the TM1-TM3 hydrophobic cluster that correlated with pore opening for potential clues on the molecular mechanism of channel gate opening. The time trajectories of dOrai across all variants showed that most of the residues in the hydrophobic stack display spatial fluctuations of small amplitude, consistent with their role as a rigid relay from TM3 to TM1. However, intriguingly, in simulations of the active variants H206Q/C (equivalent to hOrai1 H134Q/C), the M173 (hOrai1 M101) side-chain was uniquely flexible and reversibly sampled rotamers that deviated from the crystal structure (*Figure 2C,D*, *Figure 2—video 2*). We observed that the flexibility of M173 allowed it to interact with the hydrophobic gate residue, F171 (hOrai1 F99) of neighboring subunits (*Figure 2C,D*). Specifically, the M173-F171 distance often reached 4–6 Å (*Figure 2E*), sufficiently close for direct contact between the two side-chains. This finding is notable as sulfur-aromatic interactions involving Met and Phe residues are known to play distinctive roles in protein function, often by stabilizing protein structures (*Valley et al., 2012*; *Weber and Warren, 2019*). Because the M173-F171 interaction was preferentially seen only in the activating mutants and not in WT or LOF mutants (*Figure 2A–D*, *Figure 2—*

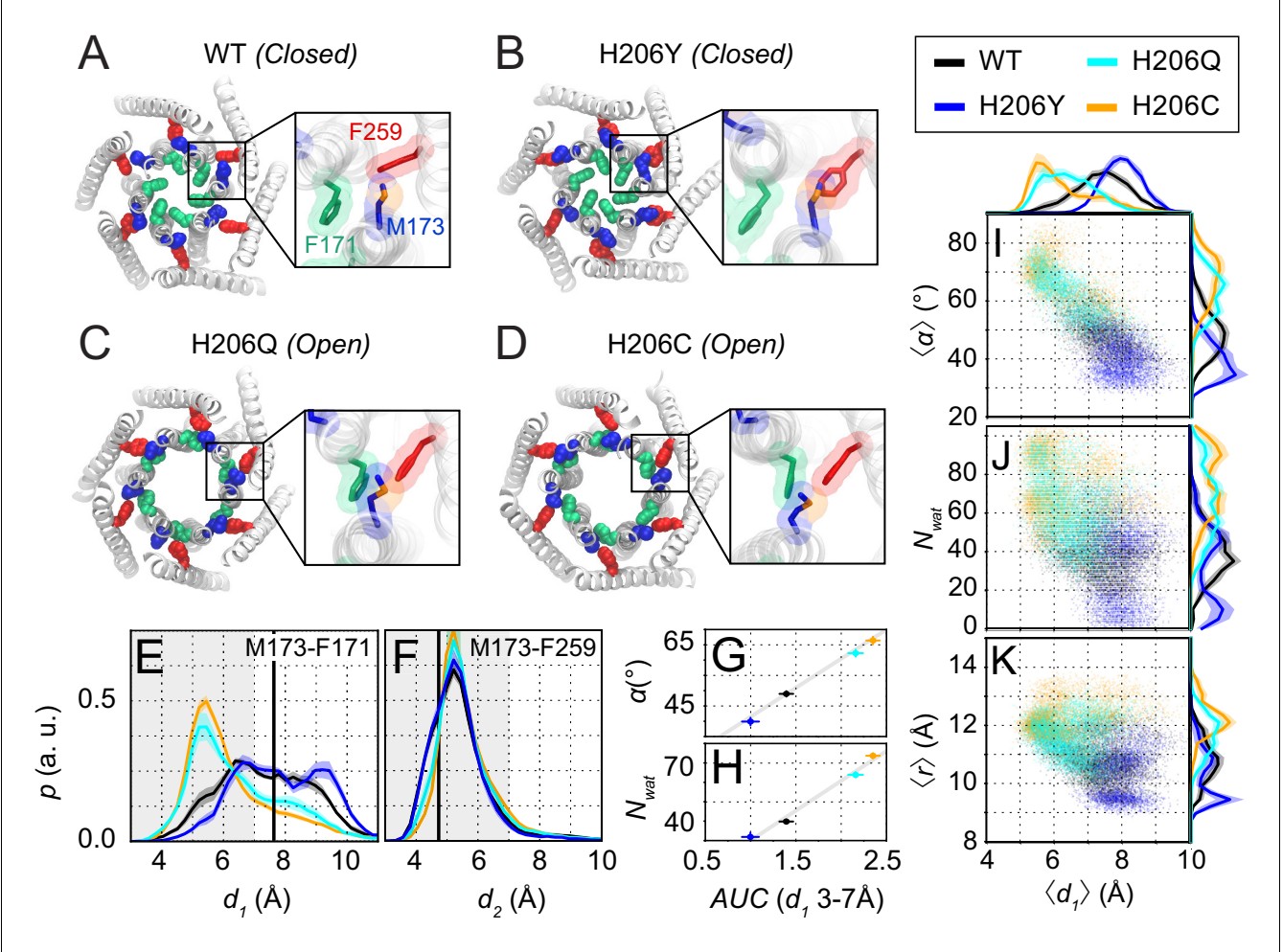

**Figure 2.** Molecular dynamics simulations reveal sulfur-aromatic interactions between M173 and F171 in open but not closed channel states. (A–D) Snapshots of WT, H206Y (hOrai1 H134Y), H206Q (hOrai1 H134Q) and H206C (hOrai1 H134C) dOrai mutants from the MD simulation runs showing positions of key residues in the TM1 and TM3 helices. F171 (hOrai1 F99, green), M173 (hOrai1 M101, blue), and F259 (hOrai1 F187, red) are represented as spheres. Insets: Enlarged views of the F171-M173-F259 locus (hOrai1 F99-M101-F187) with sulfur atoms of M173 shown in orange. (E–F) Distribution of distances between M173 with F171 ($d_1$) and M173 with F259 ($d_2$), respectively. Distances were measured from the sulfur of the methionine and the center of mass of the phenylalanine ring. The mean and standard error of mean of M173-F171 distances over simulation repeats are 7.4 ± 0.1 Å for WT, 6.4 ± 0.2 Å for H206C, 6.6 ± 0.1 Å for H206Q, and 7.9 ± 0.1 Å for H206Y. Compared to closed WT and H206Y channels, constitutively active dOrai mutants H206Q/C show a greater proportion of M173-F171 interactions within the sulfur-aromatic interaction distance of 3–7 Å. By contrast, M173-F259 distance was constant across the different dOrai variants. Black solid lines in (E) and (F) represents distances observed in the closed dOrai crystal structure 4HKR. (G–H) Area under the curve (AUC) of M173-F171 interaction distances within 7 Å in panel E plotted against pore helix rotation (G) and number of pore waters (H) (Pearson correlation coefficient of 0.98 and 0.99, respectively). (I–K) Scatter plots of the hexameric average of M173-F171 distances plotted against corresponding average pore helix rotation (I), pore hydration (J) and average pore diameter (K). The distributions of the x and y parameters are displayed on the periphery of the graphs. In general, shorter M173-F171 distances were associated with increased pore helix rotation, pore hydration and pore diameter across the Orai variants. These dynamical channel properties are anti-correlated within simulations of each system (Pearson correlation coefficient of −0.65 (WT), −0.68 (H206C), −0.72 (H206Q), and −0.17 (H206Y) for panel I). Similarly, these properties were anti-correlated across our mutational landscape (Pearson correlation coefficients of −0.79 (panel I), −0.39 (panel J), and −0.56 (panel K)). All analysis was done after the equilibration window of 100 ns.

The online version of this article includes the following video and figure supplement(s) for figure 2:

**Figure supplement 1.** Phenotypes of the loss- and gain-of-function mutants at the Orai1 H134 gating locus.

**Figure supplement 2.** Pore hydration and M173 orientation in MD simulations H206 mutants.

**Figure 2—video 1.** MD simulation trajectories of WT dOrai reveal interaction of M173 with F259.

https://elifesciences.org/articles/60751#fig2video1

**Figure 2—video 2.** MD simulation trajectories of the open mutant H206C dOrai reveal interactions of M173 with F171 and F259.

https://elifesciences.org/articles/60751#fig2video2

*videos 1* and *2*), we considered the possibility that this interaction may promote the opening of the channel gate in the activating mutants.

To investigate this hypothesis, we examined how interactions between M173 and F171 correlated with metrics of channel activity in WT, H206Y, H206Q, and H206C channels (*Figure 2G–K*). Previous studies have concluded that sulfur-aromatic interactions occur at distances less than 7 Å, with a peak distance of ≈ 5 Å (*Gómez-Tamayo et al., 2016*; *Valley et al., 2012*). Therefore, we quantified the interactions between M173 (hOrai1 M101) with the hydrophobic gate F171 (hOrai1 F99) as well as with its normal 'resting state' partner in the hydrophobic stack, F259 (hOrai1 F187 in TM3), by measuring distances between the sulfur atom on M173 and the center of mass of the aromatic rings of F171 and F259 (*Figure 2E,F*). This analysis showed that the distance between M173 and F259 oscillated between 4 and 7 Å, with an average value of 5.46 ± 0.04 Å (WT), 5.38 ± 0.05 Å (H206Y), 5.65 ± 0.06 Å (H206Q), and 5.65 ± 0.04 Å (H206C). These values indicate that the interaction of the M173 side-chain with F259 within the TM1-TM3 hydrophobic stack is relatively stable and maintained across open and closed variants (*Figure 2F*).

By contrast, the distance between M173 and F171 showed much larger variability and depended on channel activity. In closed WT and LOF H206Y (hOrai1 H134Y) channels, there was a broad distribution of F171-M173 distances from 5 Å to 10 Å (*Figure 2A,B,E*). In the activated channels H206Q and H206C, however, the distribution was markedly shifted toward shorter distances, with a peak between 4 and 7 Å, centered at 6.6 ± 0.1 Å for H206Q and 6.4 ± 0.2 Å for H206C (*Figure 2C–E*). These distances imply the presence of a stable sulfur-aromatic interaction between M173 and F171 in the open variants. Importantly, the M173-F171 interaction distance was correlated with several metrics of channel activation, including pore helix rotation, pore hydration, and pore diameter across all dOrai variants (*Figure 2G–K*), with shorter interaction distances most strongly associated with F171 rotation. The increased magnitude of counterclockwise angular rotations of F171 $C_\alpha$ in the activated mutants also correlated well with those of M173, suggesting that as the channel opens, M173 rotates along with F171 to enable it to contact F171 on the adjacent helix (*Figure 2—figure supplement 2D*). Together, these results suggest that whereas M173 interacts with F259 in both open and closed channels, it interacts with F171 only in open states. This pattern of M173 interactions seen in the closed and open mutants suggests that M173 (hOrai1 M101) acts as a bridge between F259 (hOrai1 F187) and F171 (hOrai1 F99) in the adjacent protomer to help stabilize the rotated F171 gate in the open state. The energetic contribution of the M173-F171 interaction for Orai gating will be analyzed in the Discussion section.

While the Met-Phe interactions described above may be mediated by side-chain fluctuations, rigid body motions of the TM1 backbone may also contribute to these interactions. To assess this possibility, we quantified the distances between the $C_\beta$ of M173 and the center of the F171 and F259 rings in the different H206 mutants (*Figure 2—figure supplement 2E,F*). Compared to the sulfur atom distances, $C_\beta$ measurements are likely to be driven more by pore helix rotation because the M173 $C_\beta$ is closer to the center of the TM1 helix. In WT and H206Y channels, M173 is predominantly facing TM3 as part of the TM1-TM3 hydrophobic clamp, so that $C_\beta$ is closer to F259 and farther away from F171. However, in the activated states of the H206Q/C channels, the TM1 rotation angle distribution displayed an additional peak corresponding to distances closer to F171 and farther from F259 (*Figure 2—figure supplement 2E,F*). Interestingly, analysis of the M173 $C_\beta$-F259 distances revealed clear differences between closed (WT and H206Y) and open (H206C and H206Q) channel states (*Figure 2—figure supplement 2F*). This result suggests that the TM1 helix is sufficiently mobile to display conformational alterations at the M173 $C_\beta$ position between open and closed states. By contrast, as already indicated above, no differences were seen in the distance measured from the sulfur atom to the F259 ring (*Figure 2F*), indicating that the M173 side-chain exhibits sufficient flexibility and that the sulfur atom is intrinsically drawn toward the F259 ring.

## Enhancing the M101-F99 interaction boosts F99C/M101C Orai1 channel activity

The finding that M173-F171 (hOrai1 M101-F99) interactions are augmented in MD simulations of active channel states led us to consider whether artificially forcing an interaction between M101 and F99 in hOrai1 via a metal-ion bridge can directly activate the channel. Metal-ion bridges in double cysteine mutants have been exploited in many studies of ion channel gating to probe the conformational changes underlying gating and to stabilize channels in specific states (*Holmgren et al., 1996*;

*Li et al., 2010*; *Loussouarn et al., 2001*; *McNally et al., 2012*; *Puljung and Zagotta, 2011*; *Rulísek and Vondrásek, 1998*). We therefore introduced cysteines at F99 and M101 in hOrai1 and examined the effects of applying the thiol-reactive divalent ion $Cd^{2+}$ in F99C/M101C Orai1 channels overexpressed in HEK293-H cells without STIM1 (*Figure 3A*).

F99C/M101C channels are partially open at baseline, producing non-selective CRAC currents ($V_{rev}$ = −5.4 ± 2.7 mV) in line with previous results indicating that the F99C mutation disrupts the hydrophobic barrier in the pore (*Yamashita et al., 2017*). Strikingly, $Cd^{2+}$ administration dramatically enhanced this F99C/M101C current with a slow time course over tens of seconds (*Figure 3C,E,F*). Over the 40 s application of 5 μM $Cd^{2+}$, the non-selective F99C/M101C Orai1 current increased 12-fold over the baseline current (*Figure 3C,D*). Washout of $Cd^{2+}$ by 20 mM $Ca^{2+}$ Ringer's solution caused an additional rapid increase in the current (*Figure 3C,E,F*). Previous reports have shown that application of $Cd^{2+}$ to the single F99C mutant results in strong blockade of Orai1 currents when F99C residues are oriented in a pore-facing configuration (*McNally et al., 2009*; *Yamashita et al., 2017*). Thus, we reasoned that the rapid current increase in the F99C/M101C double mutant following washout of $Cd^{2+}$ arises from the removal of $Cd^{2+}$ block by the permeating $Ca^{2+}$ ions. The $Cd^{2+}$-mediated potentiation was stable and could only be reversed by reducing agent bis(2-mercaptoethylsulfone) (BMS) (*Figure 3C,E,F*), indicating that the trapping of $Cd^{2+}$ between the introduced cysteine residues is destabilized only by directly disrupting the coordinating thiol groups.

WT and single cysteine mutants F99C and M101C did not exhibit $Cd^{2+}$-dependent current enhancement (*McNally et al., 2009*), indicating that cysteines at both positions are required for the potentiation effect. Further, when STIM1 was co-expressed, the extent of $Cd^{2+}$-induced current potentiation was dramatically reduced (*Figure 3D*). Because STIM1 stabilizes F99 is an activated state with F99 facing away from the pore and toward M101, we reasoned that the additional gating induced by a $Cd^{2+}$ bridge between F99C and M101C is diminished in the STIM1-bound state of Orai1. However, one caveat to this interpretation is that STIM1 does not activate the F99C/M101C channels strongly (I = −1.49 pA/pF at baseline versus −3.17 pA/pF following STIM1 binding). Thus, we cannot formally out that the smaller degree of STIM1-activated $Cd^{2+}$ potentiation of F99C/M101C channels is due to unknown indirect effects such as stabilization of the channel in an intermediate state where the $Cd^{2+}$-binding site is obscured. Nevertheless, the robust potentiation of F99C/M101C activity in STIM1-free channels is strong indication of the importance of M101-F99 interaction for pore opening.

Increasing the $Cd^{2+}$ dose from 5 μM to 500 μM dramatically increased the extent of channel potentiation in F99C/M101C channels, up to 150 times the baseline amplitude at 300 μM $Cd^{2+}$ (-*Figure 3E,G*). However, even at 500 μM $Cd^{2+}$ potentiation did not reach saturation. Further increases could not be quantified because $Cd^{2+}$ precipitated out of the 20 mM $Ca^{2+}$ Ringer's solution at higher concentrations. In order to distinguish whether the $Cd^{2+}$ potentiation site is close to the pore or away from it, we varied the permeant ion concentration, reasoning that a site located in or close to the pore would be highly sensitive to the concentration of permeant ions. Specifically, current potentiation is expected to decline at higher concentrations of permeant $Ca^{2+}$ ions due to a knockoff effect of $Cd^{2+}$ by $Ca^{2+}$ ions. As predicted for a $Ca^{2+}$-permeable channel, F99C/M101C channels conducted more baseline current when the external $Ca^{2+}$ concentration was raised from 20 to 110 mM $Ca^{2+}$ (*Figure 3F*). However, when applied in the 110 mM $Ca^{2+}$ solution, $Cd^{2+}$ induced significantly less current potentiation than that in 20 mM $Ca^{2+}$ solution (*Figure 3F,G*), indicating that raising the permeant ion concentration diminishes $Cd^{2+}$-induced gating. Based on this result, we conclude that $Ca^{2+}$ flux in the pore competes with $Cd^{2+}$ binding at its potentiation site and thus the $Cd^{2+}$ binding site is located close to the pore where it is influenced by permeant ions.

To test whether the gating induced by $Cd^{2+}$ is also seen with other thiol-reactive probes, we tested the effects of $Zn^{2+}$, a slightly smaller metal ion that, like $Cd^{2+}$, can also coordinate with thiol groups (*Yellen et al., 1994*). Administration of $Zn^{2+}$ also elicited potentiation of Orai1 current (*Figure 3—figure supplement 1*), suggesting that the potentiation effect is not unique to $Cd^{2+}$ but arises due to bridging of F99C to M101C via a thiol-reactive divalent ion. These results are consistent with the hypothesis that the M101-F99 interaction stabilizes the open state.

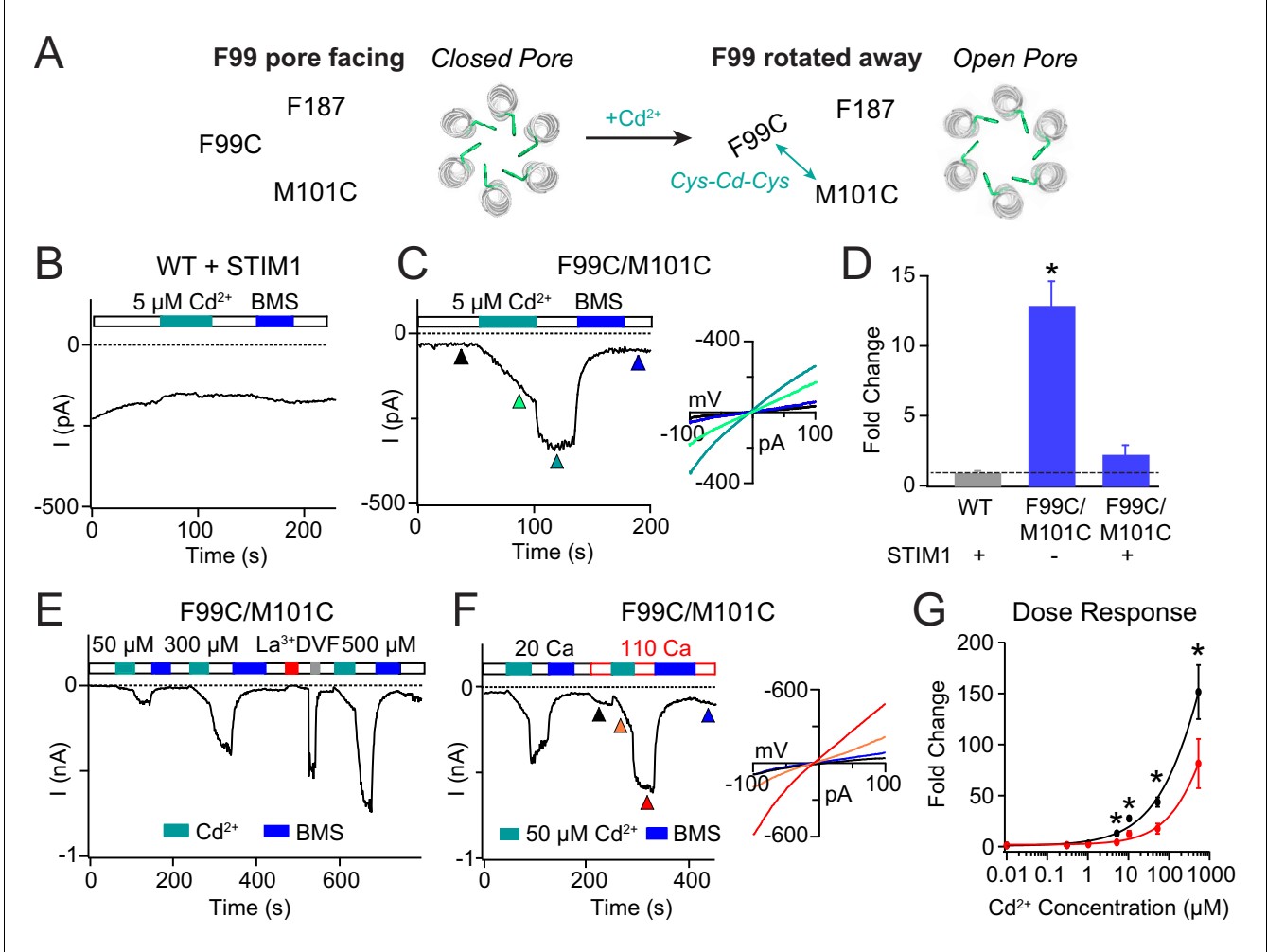

**Figure 3.** Enhancing the M101-F99 interaction via a metal-ion bridge activates F99C/M101C Orai1 in the absence of STIM1. (**A**) Schematic of proposed mechanism of action of Cd²⁺ on F99C/M101C channels. (**B**) Cd²⁺ (5 μM) has no effect on WT Orai1 channels gated by STIM1. (**C**) By contrast, the activity of the F99C/M101C Orai1 variant without STIM1 is significantly boosted by Cd²⁺ (5 μM) and can be reversed by BMS (5 mM). Inset shows the current-voltage relationships of F99C/M101C at the time points indicated by the arrowheads. (**D**) Summary of Cd²⁺ potentiation (5 μM) on F99C/M101C channels without and with STIM1. (**E**) Dose-dependence of Cd²⁺ potentiation on F99C/M101C channels (40 s of Cd²⁺ application). (**F**) Application of Cd²⁺ in 110 mM Ca²⁺-containing external solution significantly decreases the relative extent of current potentiation. Note the larger pre-Cd²⁺ baseline current amplitude in the 110 mM Ca²⁺ solution. Inset: Current-voltage relationships of F99C/M101C in 110 mM Ca²⁺ solution at the indicated time points. (**G**) Summary of the fold increase in current amplitude of F99C/M101C channels following Cd²⁺ application in 20 mM versus 110 mM Ca²⁺ external solutions. Black and red lines represent polynomial fits to the data to enable visualization of the overall trends. Less relative potentiation is seen in 110 mM Ca²⁺ solution, suggesting that permeating Ca²⁺ can compete with Cd²⁺ for a binding site within the pore. Values are mean ± S.E.M. N = 4–11 cells *p<0.05 by Student's T-test. Numerical data for this figure can be found in *Figure 3—source data 1*.

The online version of this article includes the following source data and figure supplement(s) for figure 3:

**Source data 1.** Numerical data for cadmium-mediated current potentiation in F99C/M101C Orai1.

**Figure supplement 1.** Cd²⁺-mediated Orai1 current potentiation and inhibition is replicated by Zn²⁺.

## Stabilizing the M101-F187 interaction promotes pore closure in M101C/F187C Orai1 channels

If stabilizing a M101-F99 interaction augments channel activation, does facilitating the resting state interaction between M101 and F187 (as seen in the closed channel crystal structure) favor the closed state? We tested this idea using the hOrai1 M101C/F187C mutant. We have previously shown that Orai1 F187C is a Ca²⁺-selective, constitutively active mutant (*Yeung et al., 2018*). Similarly, the M101C/F187C double cysteine mutant was also constitutively active and highly Ca²⁺ selective (*Figure 4B*). Strikingly, when 5 μM Cd²⁺ was applied to M101C/F187C, the current was almost

completely inhibited (*Figure 4B*). This current decrease was uniform across the voltage ramp ($V_{rev}$ = 41.0 ± 6.3 mV without $Cd^{2+}$ and $V_{rev}$ = 35.8 ± 8.9 mV with 5 µM $Cd^{2+}$; p=0.65), which suggests that it arises from channel inhibition rather than pore block. In addition, the reversibility of the effect by BMS is consistent with that of a cysteine-mediated mechanism (*Figure 4B*). The rapid $Cd^{2+}$-induced inhibition was only seen in the double Cys mutants. Neither the single F187C nor double M101A/F187C and M101C/H134S mutants showed any effect with $Cd^{2+}$ (*Figure 4C*), indicating that cysteines are required at both F187 and M101 for inducing channel closure.

In contrast to the strong dependence on the permeant ion concentration seen for the $Cd^{2+}$-mediated potentiation of F99C/M101C channels, $Cd^{2+}$-induced channel closure in the M101C/F187C mutant was largely insensitive to the permeant ion concentration. Specifically, $Cd^{2+}$ blockade in 110 mM $Ca^{2+}$ Ringer's was not notably different than that seen in 20 mM $Ca^{2+}$ solution (*Figure 4E,F*), suggesting that the inhibition site is likely not in the pore. This conclusion is also supported by the strikingly different $Cd^{2+}$ sensitivities of $Cd^{2+}$ inhibition on M101C/F187C channels versus potentiation in F99C/M101C channels. Whereas inhibition in M101C/F187C channels quickly reached saturation with increasing $Cd^{2+}$ with an apparent $K_d$ of ~0.3 µM (*Figure 4F*), potentiation in the F99C/M101C channels did not reach saturation even at 500 µM $Cd^{2+}$ (*Figure 3G*). The latter effect is likely due to a knockoff effect of the permeant $Ca^{2+}$ ions on $Cd^{2+}$ occupancy at the potentiation site between F99C and M101C. Application of $Zn^{2+}$ also induced channel closure in M101C/F187C channels, suggesting that the inhibition effect is generalizable to other thiol-reactive metals (*Figure 3—figure supplement 1*). We conclude that stabilizing the resting interaction between M101 and F187 evokes pore closure, presumably by releasing the F99 gate into its pore-facing configuration (*Figure 4A*).

## M101 is essential for STIM1-mediated Orai1 channel activation

If the sulfur-aromatic interaction is critical for gate opening, then mutations of M101 to other amino acids that cannot support sulfur-aromatic interactions should disrupt channel function. Consistent with this hypothesis, nearly every substitution that we tested, including M101G/A/S/T/C/V/L/I, yielded LOF channels that lost gating by STIM1 (*Figure 5A,B*). Notably, mutations of M101 to Leu or Ile, which have comparable or even greater hydrophobicity as the native Met, also produced channels that could not be activated by STIM1 (*Figure 5A,B*). The plasma membrane fluorescence of Orai1-YFP was similar across the mutants, indicating that reduced channel expression is not the cause of the smaller currents. Moreover, with the exception of some polar and charged substitutions (M101Q/N/D/K/E), no difference was seen in co-localization of the YFP-tagged Orai1 mutants with CFP-CAD nor in E-FRET between the two proteins (*Figure 5—figure supplement 1*), indicating that loss of gating was not due to gross changes in the ability of the Orai1 variants to interact with STIM1. Introduction of the M101L mutation into the constitutively active H134S and constitutively conducting V102C mutants also attenuated currents in these mutants (*Figure 5—figure supplement 2*). This latter result in V102C is not entirely unexpected since previous studies have indicated that the V102C mutant also shows greater spontaneous counter-clockwise fluctuations of the F99 gate region compared to WT channels, consistent with models showing that gate opening and pore hydration are intimately coupled in CRAC channels (*Yamashita et al., 2017*; *Yeung et al., 2018*).

The one exception to the LOF effect induced by mutations at M101 was the Phe mutant, which surprisingly produced a GOF phenotype. M101F was not only normally gated by STIM1 (*Figure 5A*), but was also modestly active and $Ca^{2+}$-selective ($V_{rev}$ = 44.9 ± 5.7 mV) at rest in the absence of STIM1 (*Figure 5C*). Interestingly, the M101F substitution was able to confer channel activity to LOF mutant H134W (*Figure 5—figure supplement 2*) which cannot be gated by STIM1 (*Yeung et al., 2018*), suggesting that the effect of M101F is directly on the F99 gate. As will be discussed further below, we postulate that the constitutive activity of M101F likely arises from interactions of the introduced Phe side-chain with F99 and F187 created by the introduced aromatic ring at M101. Collectively, these M101 mutants demonstrate that hydrophobicity alone is not enough for channel function and reaffirm the conclusion that the sulfur-containing Met residue at position 101 mediates a specialized function in Orai1 gating in addition to its role in the hydrophobic cluster.

In contrast to the LOF Orai1 phenotypes of mutations at M101, mutation of F187 to smaller residues including Gly, Ala, Ser, and Cys produced GOF channels with inwardly rectifying CRAC channel-like current-voltage relationships. Large and hydrophobic F187 substitutions including Tyr and Trp, on the other hand, remained closed in the absence of STIM1 (*Figure 5—figure supplement 3*).

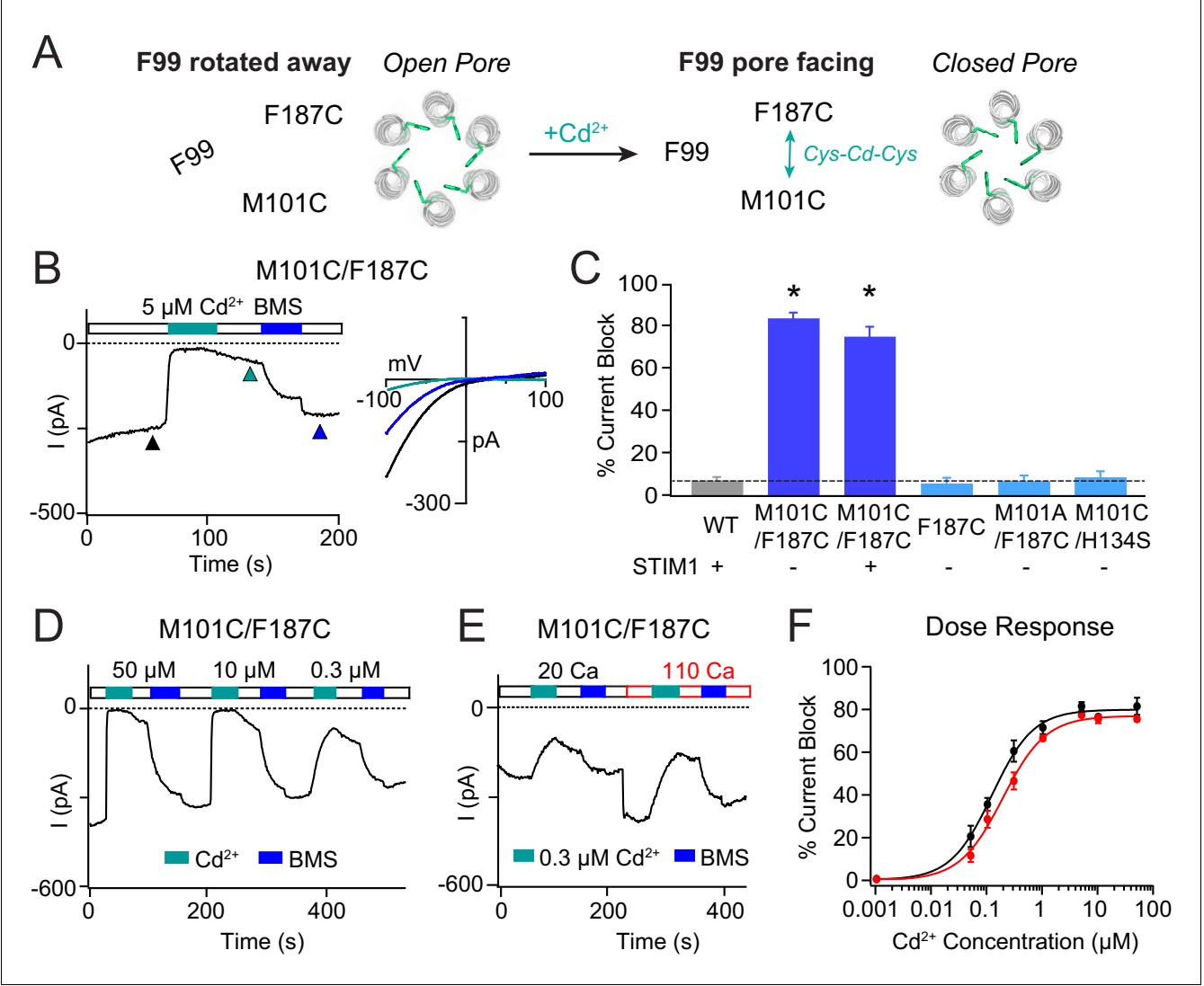

**Figure 4.** Stabilizing the M101-F187 interaction via a metal-ion bridge promotes pore closure. (A) Schematic of proposed mechanism of action of $Cd^{2+}$ on M101C/F187C channels. The addition of $Cd^{2+}$ induces a bridge between M101C and F187C which releases the F99 gate returning to the 'closed' orientation. (B) $Cd^{2+}$ (5 μM) strongly inhibits the current of M101C/F187C channels, which can be reversed by BMS (5 mM). Inset: Current-voltage relationships of M101C/F187C at the time points indicated by the arrowheads. (C) Summary of $Cd^{2+}$ inhibition (5 μM) on M101C/F187C channels without and with STIM1. Double mutants with only one cysteine at positions M101C or F187C do not exhibit block by $Cd^{2+}$, indicating that the coordination of $Cd^{2+}$ requires cysteines at both residues. (D) M101C/F187C channels display a dose-dependent increase in current inhibition (40 s of $Cd^{2+}$ application). (E) Application of $Cd^{2+}$ in 110 mM $Ca^{2+}$-containing external solution does not notably affect the extent of current inhibition by $Cd^{2+}$. (F) Summary of the fold increase in current amplitude of M101C/F187C channels following $Cd^{2+}$ application in 20 mM versus 110 mM $Ca^{2+}$ external solutions. The lack of difference in effect suggests that the $Cd^{2+}$ binding site is not within the pore, and instead at the TM1-TM3 interface as implied by the crystal structure. Values are mean ± S.E.M. N = 4–8 cells for each point; *p<0.05 by Student's T-test. Numerical data for this figure can be found in *Figure 4—source data 1*.

The online version of this article includes the following source data for figure 4:

**Source data 1.** Numerical data for cadmium-mediated current inhibition in M101C/F187C Orai1.

Based on these results, we conclude that the large, hydrophobic side-chain of F187 on TM3 is required for stabilizing the closed state of the channel. However, because M101C channels are not constitutively open, we surmise that F187 may prevent spontaneous pore opening through mechanisms beyond its interaction with M101, potentially involving other residues of the TM1-TM3 hydrophobic stack. Without additional these interactions, the pore transitions into a $Ca^{2+}$-selective state that is similar to the one evoked by STIM1 binding.

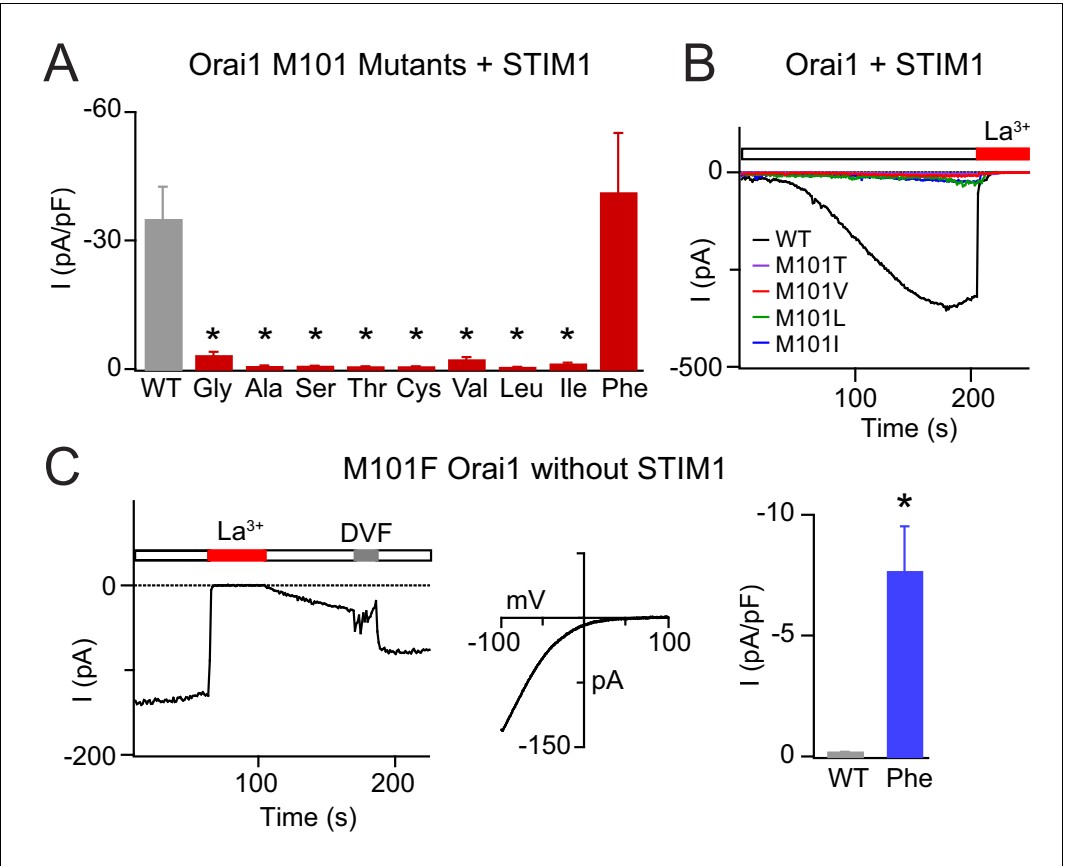

**Figure 5.** Mutations at M101 expected to disrupt M101-F99 interactions abrogate Orai1 gating. (**A**) Current densities of Orai1 M101 mutants in the presence of STIM1. The majority of M101 mutations, even those to large residues similar in hydrophobicity to the native Met, abolish Orai1 activation by STIM1. (**B**) Time course traces of LOF M101 mutants with co-expressed with STIM1. Unlike WT Orai1 channels, these mutants do not conduct current after store-depletion by 8 mM BAPTA in the internal solution. (**C**) M101F evokes a GOF effect. This mutant is constitutively active even in the absence of STIM1. The constitutively active M101F variant is blocked by $La^{3+}$ (150 µM) and its current-voltage relationship indicates a highly $Ca^{2+}$-selective current. Values are mean ± S.E.M. N = 4–9 cells *p<0.05 by Student's T-test. Numerical data for this figure can be found in *Figure 5—source data 1*. The online version of this article includes the following source data and figure supplement(s) for figure 5:

**Source data 1.** Numerical data for M101 Orai1 mutants.
**Figure supplement 1.** The LOF M101 mutants retain normal levels of binding to STIM1.
**Figure supplement 2.** Additional analysis of M101 mutations.
**Figure supplement 3.** F187 is required for stabilizing the closed channel state.

## Loss of the gate latch prevents channel opening by stabilizing the gate in its closed position

The experimental results showing that loss of the sulfur-aromatic gate latch compromises channel gating led us to hypothesize that M101 LOF mutations stabilize the F99 gate in the closed pore-facing orientation, resulting in diminished pore hydration at the channel gate. To address this hypothesis, we performed MD simulations of dOrai M173L (LOF hOrai1 M101L) and dOrai M173F (GOF hOrai1 M101F) to investigate the effects of the mutations on pore hydration and orientation of the F99 gate. These simulations revealed that the M173L channel displayed significantly smaller spontaneous counter-clockwise pore helix rotations than the WT channel (*Figure 6D*), with the angular position of F171 shifted by 10 ± 1˚ for M173L channels and 17 ± 1˚ for WT channels as measured with respect to the crystallographic orientation of the F171 side-chain in our coordinate system (30.6˚, [PDB ID:4HKR]; *Hou et al., 2012*). The decrease in pore helix rotation of M173L was accompanied by markedly lower hydration of the hydrophobic stretch of the pore (*Figure 6B*). By contrast,

M173F channels, which are modestly active in the absence of STIM1 (*Figure 5C*), showed a statistically significant increase in the extent of pore helix rotation relative to WT (*Figure 6D*) and a similar level of pore hydration as WT channels (*Figure 6B*).

Examination of the simulation trajectories showed that the introduced Leu, which is both shorter and more hydrophobic than the native Met, spends the majority of the simulation time in the hydrophobic cluster at the TM1-TM3 interface and away from F171, suggesting that it cannot interact with F171 to facilitate pore opening (*Figure 6—video 1*). Quantification of the M173L-F171 distance as measured from the $C_\beta$ atom of M173 indicated a subtle but clear shift towards larger distances in this variant. This shift was accompanied by an increase in the interaction propensity between M173L and F259 (*Figure 6E,F* and *Figure 6—video 1*). By contrast, in simulations of M173F channels, we observed that the Phe side-chain, which is shorter and less flexible than the native Met at position 173, results in a bridge between F171 and the TM3 residue F259, causing F171 to be displaced away from the pore and increasing pore helix rotation (*Figure 6—video 2*). We conclude that the M173L mutation stabilizes F99 in a pore-facing configuration, and that the resulting decrease in pore hydration relative to WT channels in the hydrophobic stretch contributes to the LOF phenotype of the M101L mutant.

## Discussion

Ion channel gates come in a variety of shapes and sizes, including bulky amino acids that cause steric blockade, reversible salt bridges that form electrostatic barriers, and hydrophobic residues that impede pore hydration. Recent progress in membrane protein purification, high-resolution structural biology techniques, and molecular dynamics studies has permitted detailed examination of the structural and energetic landscape within channel pores. We now know that many ion channels operate, at least in part, by hydrophobic gating, wherein energetically unfavorable interactions between water molecules and nonpolar pore-lining residues lead to a dewetted stretch in the pore that presents an energetic barrier to ion conduction (*Aryal et al., 2015*).

The hydrophobic gate of store-operated Orai1 channels is formed by two rings of pore-facing nonpolar residues, V102 and F99, which together preclude ion flow in the resting state (*McNally et al., 2012*; *Yamashita et al., 2017*). In this study, we identified a unique feature of Orai1 channels, the M101 gate latch, which facilitates displacement of the F99 gate to allow ion permeation. Sulfur-aromatic interactions are prevalent across diverse groups of proteins and have been shown to be essential in stabilizing protein folding and facilitating oxidation-dependent conformational changes (*Gómez-Tamayo et al., 2016*; *Valley et al., 2012*; *Weber and Warren, 2019*). More recently, a systematic survey of all protein structures available in the Protein Data Bank reported that up to 40% of all proteins involved a methionine bridging two aromatic residues in Aro-Met-Aro formations, albeit with less well-defined functional roles than simple Met-aromatic interactions (*Weber and Warren, 2019*). To our knowledge, the methionine gate latch in CRAC channels is the first reported example of a Met-aromatic interaction involved in ion channel gating.

MD simulations indicated that, contrary to our previous predictions of its role in a conformationally constrained hydrophobic clamp (*Yeung et al., 2018*), M101 retains a considerable amount of flexibility. Most notably, direct M101-F99 contact was strongly correlated with metrics of pore opening (*Figure 2*), suggesting that M101 acts as a bridge between the F99 gate and an F187 'anchor' on TM3 to keep the channel gate open. We validated this concept experimentally by introducing double cysteine mutants that can be bridged by $Cd^{2+}$. In the F99C/M101C double mutant, formation of a $Cd^{2+}$ bridge between the cysteines at F99 and M101 resulted in robust activation of Orai1 current (*Figure 3*). This effect is reminiscent of $Cd^{2+}$ bridging between cysteines introduced at G98 and F99, which also potentiates Orai1 current by tethering F99C in a deflected state away from the pore (*Yamashita et al., 2017*). The opposite is true for the M101C/F187C double mutant, where $Cd^{2+}$ disrupts the constitutive activation of the mutant, presumably by forming a bridge between M101C and F187C. The formation of the M101C-$Cd^{2+}$-F187C bond likely destabilizes the open state of the F99 gate, freeing F99, and causing it to revert into its closed pore-facing configuration to close the channel (*Figure 4*).

Consistent with an essential role for the M101 gate latch in opening the gate, none of the tested mutations at M101 supported STIM1-mediated Orai1 activation. Even the substitutions that are most similar to the native Met (e.g. Leu, Ile, Cys) in their hydrophobicity and which preserve the

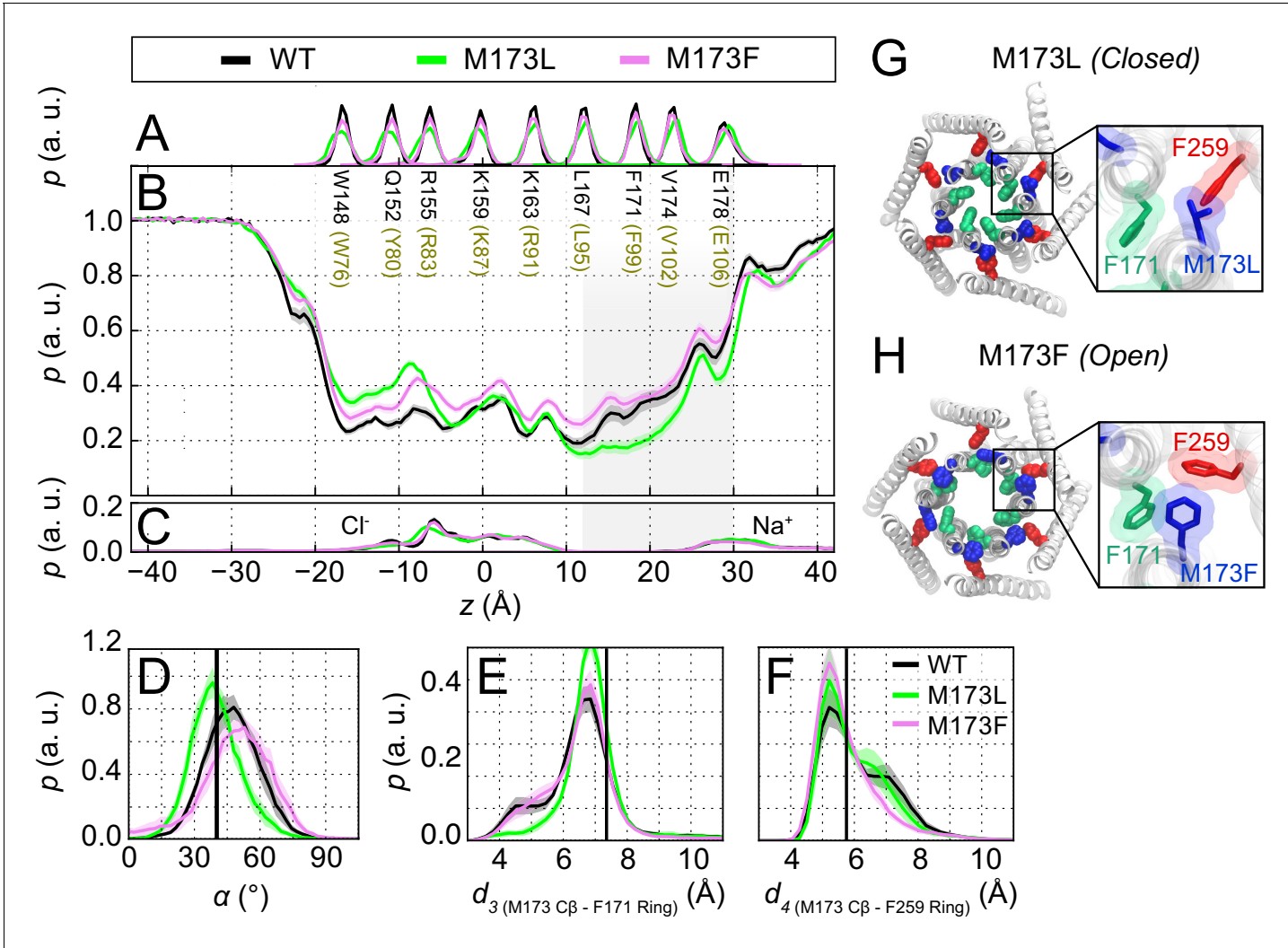

**Figure 6.** The loss-of-function M173L mutation decreases pore hydration and stabilizes closure of the hydrophobic gate. (A) Relative probability distributions of the axial position of $C_\alpha$ atoms for all pore-lining residues (hOrai1 numbering in brown). Average distribution of water oxygen atoms (B) and $Na^+$ and $Cl^-$ ions (C) along the pore axis. The P values from a two-sided Welch's t-test of mean water oxygen count at z = 15 Å across all simulation repeats indicate a significant difference with respect to WT for M173L ($1.2 \times 10^{-2}$, p<0.05) but not for M173F ($1.1 \times 10^{-1}$). (D) Relative distributions of the radial angle of residue 171 defined as the angle between the pore axis, the center of mass of the two helical turns centered at residue 171, and the $C_\alpha$ atom of residue 171 in the different mutants. The mean and standard error of mean of F171 radial angle over simulation repeats are 41 ± 1 ° for M173L and 48 ± 1 ° for M173F. The P values from a two-sided Welch's t-test for each system indicate significant differences (p<0.05) between the mean of these distributions with respect to WT simulations ($6.1 \times 10^{-24}$, $5.2 \times 10^{-3}$). The black solid line represents the angle observed in the closed dOrai crystal structure 4HKR. (E–F) Distribution of distances between M173 with F171 ($d_3$) and M173 with F259 ($d_4$), respectively. Distances were measured from the $C_\beta$ of the methionine and the center of mass of the phenylalanine ring. The mean and standard error of mean of M173-F171 distances over simulation repeats are 6.5 ± 0.1 Å for WT, 6.9 ± 0.1 Å for M173L, and 6.6 ± 0.1 Å for M173F. In (F), M173-F259 distances over simulation repeats are 6.1 ± 0.1 Å for WT, 6.0 ± 0.1 Å for M173L, and 5.8 ± 0.1 Å for M173F. Black solid lines in (A) and (B) represent distances observed in the crystal structure of closed dOrai ($d_3$ = 7.3 Å and $d_4$ = 5.8 Å). (G) Snapshot of the positions of M173L, F171, and F259 in the closed M173L mutant. M173L decreases pore hydration and pore helix rotation, thereby evoking closure of the F171 hydrophobic gate. (H) Snapshot of the positions of M173F, F171, and F259 in the constitutively active M173F mutant. Only TM1 and TM3 helices are shown in (G–H) for simplicity.

The online version of this article includes the following video(s) for figure 6:

**Figure 6—video 1.** MD simulation trajectory of the LOF mutant M173L dOrai.

https://elifesciences.org/articles/60751#fig6video1

**Figure 6—video 2.** MD simulation trajectory of the GOF mutant M173F dOrai.

https://elifesciences.org/articles/60751#fig6video2

ability of Orai1 to bind STIM1 failed to be activated by co-expressed STIM1 (*Figure 5A*). These findings indicate that a combination of hydrophobicity and the presence of the sulfur atom in Met is necessary for the M101 latch function. The only exception to the loss of channel activity was the M101F mutant, which elicited a modest GOF phenotype. We speculate that the gate opening of this mutant may arise due to direct edge-to-face or face-to-face π-π stacking interactions among the three aromatic rings (F99, M101F, F187) as suggested by the simulations (*Figure 6H*) to promote displacement of the gate.

M101 is endowed with a unique combination of features that allow it to fulfill its specialized role as a latch for stabilizing the open state of the gate. First, M101 is well situated on the non-pore facing surface of TM1 towards TM3 in closed channels and is oriented towards F99 of a neighboring subunit when TM1 rotates counterclockwise and the pore dilates. Second, Met is longer and more flexible than Leu or Ile, allowing it to interact with both F187 on TM3 and F99 in the pore. Third, and most importantly, the sulfur atom in methionine provides an additional energetic contribution that is stronger than a purely hydrophobic interaction with F99 (*Reid et al., 1985*; *Valley et al., 2012*). In solution, the energy associated with each Met-Phe bond in *ab initio* simulations using dimethyl sulfide and benzene as proxies of sulfur-aromatic interactions is estimated to be 2.0–2.9 kcal/mol (*Gómez-Tamayo et al., 2016*), providing an upper bound to the free energy of the interaction. The true free energy of the M101-F99 association however, is likely to be significantly lower due to constraints within the Orai1 protein environment. Moreover, in the proposed gating mechanism, it is the *difference* in free energy between the contacts formed in open and closed states that is relevant for the gating transition.

To better estimate the contribution of the Met-Phe contacts to the gating equilibrium in Orai1, we exploited the fact that in simulations of dOrai, the open and closed channel states sample both contact and non-contact separations. The analysis in *Figure 2E* shows four distributions of M173-F171 distances, of two open (H206Q and H206C) and two closed channel variants (WT and H206Y). Assuming that each of these distributions approaches equilibrium, we can convert the distance distributions to potential of mean-force profiles (*Figure 7B*) using a relationship derived from the Boltzmann equation (see Materials and methods). In the open state modeled by H206Q/C channels, this analysis indicates a free energy minimum at 5–6 Å that is associated with the sulfur-aromatic interaction (*Figure 7B*).

The energetic contribution of the interaction to channel opening (*Figure 7C*) can be quantified from the relative probability distributions of the M173-F171 distances ($d_1$) in closed and open states. Using a distance of $d = 7$ Å as the cutoff (*Gómez-Tamayo et al., 2016*; *Valley et al., 2012*), we transformed these continuous probability profiles into discrete states corresponding to contact versus non-contact Met-Phe states. For the closed and open channels modeled by the above variants, the relative occupancy of the contact and non-contact states can be estimated from the ratio of the cumulative probabilities above and below the 7 Å cutoff. This ratio is ~0.8 for WT channels, ~0.5 for H206Y channels, ~2.2 for H206Q channels, and ~3 for H206C channels. Based on these ratios, the relative preference ($P_{contact}/P_{non-contact}$) for Met-Phe interactions is approximately 0.8:1 in the closed (WT) state and 3:1 in the open (H206C) state.

Because free energy is conserved in the thermodynamic cycle in *Figure 7C*, it follows that:

$$\Delta G_1 + \Delta G_4 - \Delta G_2 - \Delta G_3 = 0$$

where $\Delta G_1$, $\Delta G_2$, $\Delta G_3$, and $\Delta G_4$ are the free energy changes for the transitions defined in *Figure 7C*. Rearranging the terms leads to:

$$\Delta G_2 - \Delta G_1 = \Delta G_4 - \Delta G_3$$

where $\Delta G_2 - \Delta G_1$ corresponds to the relative stabilization of the contact state upon channel opening, and $\Delta G_4 - \Delta G_3$ corresponds to the relative stabilization of the open state upon making contact. $\Delta G_1$ and $\Delta G_2$ in *Figure 7C* can be calculated from the relationship:

$$\Delta G = -RTln\left(\frac{P_{contact}}{P_{non-contact}}\right)$$

where $P_{contact}$ and $P_{non-contact}$ are the probabilities of the Met-Phe distances being smaller and larger than the 7 Å cutoff, respectively. Substituting values obtained in the previous analysis, we get 0.13 kcal/mol for $\Delta G_1$ and $-0.65$ kcal/mol for $\Delta G_2$. Therefore, the $\Delta\Delta G$ for stabilization of the open state

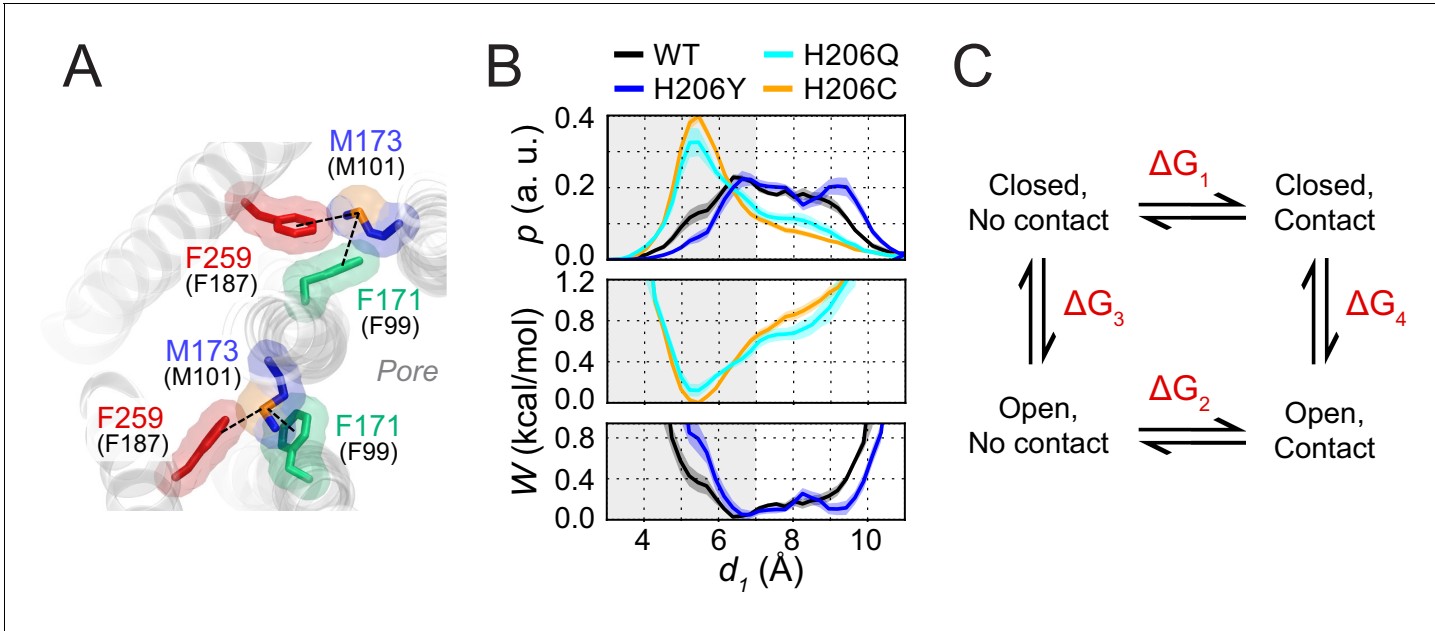

**Figure 7.** Free energy analysis of M173-F171 interactions. (**A**) A snapshot of the H206C dOrai GOF mutant showing two subunits in which the M173 latch is engaged with the F171 hydrophobic gate and F259 to evoke gate opening. hOrai1 numbering is shown in parentheses. (**B**) Top: M173-F171 distances for the various H206 mutants replotted from *Figure 2E*. The gray shaded area corresponding to $d_1 < 7$ Å represents the range of distances used to infer the occurrence of Met-Phe contact. Middle and bottom: The potential of mean-force (*W*) for the M173-F171 distance for the mutants simulated as indicated in *Figure 2*. (**C**) Schematic of thermodynamic cycle associated with the M173-F171 interaction in closed and open channel states.

by the gate latch contact is ~0.8 kcal/mol, a value that is commensurate with the magnitude of thermal energy at physiological temperatures. Assuming that the contributions of the individual Met-Phe interactions are additive, this analysis implies that the six Met-Phe interactions contribute approximately ~5 kcal/mol per channel. However, the 0.8 kcal/mol estimate could reflect in part cooperative interactions between multiple gate latch pairs rather than individual interactions, so that the actual energetic contribution to pore opening may be lower than this 5 kcal/mol estimate.

How do these estimates compare to the overall free energy change required for Orai1 channel activation? Although we currently do not know the total free energy change associated with Orai1 gating, comparison with the estimated activation energies for the Shaker potassium channel (14 kcal/mol) (*Chowdhury and Chanda, 2012*) and the BK channel (24 kcal/mol) (*Chowdhury and Chanda, 2013*) suggests that this estimate of the M101-F99 interaction energy may be critical for the overall Orai1 gating process. Furthermore, because F99 is on the face of TM1 opposite to M101, the formation of the proposed latch interactions between neighboring TM1 helices would be expected to promote correlated TM1 helix rotations to induce cooperative opening of the channel gate. These considerations indicate that M101 is well positioned, with the suitable length and added sulfur group, to interact with F187 and F99 act as an effective gate latch to facilitate opening and closing of the F99 channel gate.

From a functional standpoint, the finding M101 that promotes pore opening by stabilizing the channel gate into the open configuration likely has important implications for CRAC channel gating. Activation of CRAC channels occurs in a highly nonlinear manner with respect to STIM1 binding, resulting in abrupt switching of closed channels from a long-lasting silent state into a high open probability ($P_O$) channel state (*Prakriya and Lewis, 2006*; *Yen and Lewis, 2018*). The M101-F99 interaction allows for conformational changes in one TM1 subunit to be transmitted to a neighboring subunit because the M101 gate latch stabilizes the rotated state of the neighboring F99 while also keeping its own TM1 in a rotated conformation. This synergistic interaction could promote cooperative pore opening across the entire channel once one or more subunits are flipped into the active state. Once the pore is opened, the M101-F99 interaction can help maintain Orai1 channel's characteristic high-$P_O$ state governing the induction of downstream cellular pathways.

# Materials and methods

**Key resources table**

| Reagent type (species) or resource | Designation | Source or reference | Identifiers | Additional information |
|---|---|---|---|---|
| Cell line (*Homo-sapiens*) | HEK293-H | Thermo Fisher Scientific | 11631017 | RRID:CVCL_6643 |
| Commercial assay or kit | QuikChange II XL Site-Directed Mutagenesis Kit | Agilent | 200522 | |
| Transfected construct (human) | Orai1-YFP | Clontech *Navarro-Borelly et al., 2008* | | |
| Transfected construct (human) | mCherry-STIM1 | Richard Lewis (Stanford) | | |
| Transfected construct (human) | CFP-CAD | Richard Lewis (Stanford) | | |
| Chemical compound, drug | Lipofectamine 2000 | Thermo Fisher Scientific | 11668019 | |
| Chemical compound, drug | cadmium chloride | Sigma-Aldrich | 202908 | |
| Chemical compound, drug | BMS, bis (2-mercaptoethylsulfone) | Calbiochem | 145626-87-5 | |
| Sequenced-based reagent | mutagenesis primers for Orai1 F99C | IDT *Yamashita et al., 2017* | | accatggcgcagccggagagcagagcc ggctctgctctccggctgcgccatggt |
| Sequenced-based reagent | mutagenesis primers for Orai1 M101A | IDT | This paper | ccaccattgccaccgcggcgaagccggaga tctccggcttcgccgcggtggcaatggtgg |
| Sequenced-based reagent | mutagenesis primers for Orai1 M101C | IDT *McNally et al., 2009* | | ctccaccattgccacgcaggcgaagccggagag ctctccggcttcgcctgcgtggcaatggtggag |
| Sequenced-based reagent | mutagenesis primers for Orai1 M101F | IDT | This paper | ccaccattgccacgaaggcgaagccggag ctccggcttcgccttcgtggcaatggtgg |
| Sequenced-based reagent | mutagenesis primers for Orai1 M101G | IDT | This paper | ccaccattgccaccccggcgaagccggaga tctccggcttcgccgggggtggcaatggtgg |
| Sequenced-based reagent | mutagenesis primers for Orai1 M101I | IDT | This paper | accattgccactatggcgaagccggagag ctctccggcttcgccatagtggcaatggt |
| Sequenced-based reagent | mutagenesis primers for Orai1 M101L | IDT | This paper | accattgccaccaaggcgaagccggag ctccggcttcgccttggtggcaatggt |
| Sequenced-based reagent | mutagenesis primers for Orai1 M101S | IDT | This paper | tccaccattgccacgctggcgaagccggag ctccggcttcgccagcgtggcaatggtgga |
| Sequenced-based reagent | mutagenesis primers for Orai1 M101T | IDT | This paper | ccattgccaccgtggcgaagccggag ctccggcttcgccacggtggcaatgg |
| Sequenced-based reagent | mutagenesis primers for Orai1 M101V | IDT | This paper | accattgccaccacggcgaagccggag ctccggcttcgccgtggtggcaatggt |
| Sequenced-based reagent | mutagenesis primers for Orai1 V102C | IDT *McNally et al., 2012* | | gcacctccaccattgcgcacatggcgaagccggag ctccggcttcgccatgtgcgcaatggtggaggtgc |
| Sequenced-based reagent | mutagenesis primers for Orai1 H134C | IDT *Yeung et al., 2018* | | catgagcgcaaacaggcacacagccaccagcact agtgctggtggctgtgtgcctgtttgcgctcatg |
| Sequenced-based reagent | mutagenesis primers for Orai1 H134Q | IDT *Yeung et al., 2018* | | atgagcgcaaacagctgcacagccaccag ctggtggctgtgcagctgtttgcgctcat |
| Sequenced-based reagent | mutagenesis primers for Orai1 H134S | IDT *Yeung et al., 2018* | | catgagcgcaaacaggctcacagccaccagcact agtgctggtggctgtgagcctgtttgcgctcatg |
| Sequenced-based reagent | mutagenesis primers for Orai1 H134W | IDT *Yeung et al., 2018* | | gatcatgagcgcaaacagccacacagccaccagcactgt acagtgctggtggctgtgtggctgtttgcgctcatgatc |
| Sequenced-based reagent | mutagenesis primers for Orai1 H134Y | IDT *Yeung et al., 2018* | | atcatgagcgcaaacagatacacagccaccagcactg cagtgctggtggctgtgtatctgtttgcgctcatgat |
| Sequenced-based reagent | mutagenesis primers for Orai1 F187A | IDT *Yeung et al., 2018* | | ccacctcagctagggcgagcagcgtgccga tcggcacgctgctcgccctagctgaggtgg |

*Continued on next page*

*Continued*

| Reagent type (species) or resource | Designation | Source or reference | Identifiers | Additional information |
|---|---|---|---|---|
| Sequenced-based reagent | mutagenesis primers for Orai1 F187C | IDT *Yeung et al., 2018* | | cctcagctaggcagagcagcgtgccg cggcacgctgctctgcctagctgagg |
| Sequenced-based reagent | mutagenesis primers for Orai1 F187G | IDT | This paper | ccacctcagctaggccgagcagcgtgccga tcggcacgctgctcggcctagctgaggtgg |
| Sequenced-based reagent | mutagenesis primers for Orai1 F187L | IDT | This paper | accacctcagctagtaagagcagcgtgcc ggcacgctgctcttactagctgaggtggt |
| Sequenced-based reagent | mutagenesis primers for Orai1 F187S | IDT | This paper | ccacctcagctaggctgagcagcgtgccga tcggcacgctgctcagcctagctgaggtgg |
| Sequenced-based reagent | mutagenesis primers for Orai1 F187W | IDT | This paper | accacctcagctagccagagcagcgtgccg cggcacgctgctctggctagctgaggtggt |
| Sequenced-based reagent | mutagenesis primers for Orai1 F187Y | IDT | This paper | caccacctcagctagatagagcagcgtgccga tcggcacgctgctctatctagctgaggtggtg |
| Sequenced-based reagent | mutagenesis primers for Orai1 L273D | IDT *Li et al., 2011* | | ccgccagctcgttgtcctcctggaactgtc gacagttccaggaggacaacgagctggcgg |

## Cells

HEK293-H cells (Thermo Fisher Scientific) were maintained in suspension at 37°C with 5% $CO_2$ in CD293 medium supplemented with 4 mM GlutaMAX (Invitrogen). The HEK293 cell line is a permanent line established from primary embryonic human kidney and transformed with sheared human adenovirus type 5 DNA. The E1A adenovirus gene is expressed in these cells to optimize protein production. HEK293-H cells were cloned from the original 293 cell line and adapted to CD293 serum-free medium for growth in suspension. Cell line identity has been authenticated by Thermo-Fisher Scientific, and cells were tested negative for mycoplasma by qPCR detection assay. For imaging and electrophysiology, cells were plated onto poly-L-lysine coated coverslips one day before transfection and grown in a medium containing 44% DMEM (Corning), 44% Ham's F12 (Corning), 10% fetal bovine serum (HyClone), 2 mM glutamine, 50 U/ml penicillin and 50 µg/ml streptomycin.

## Plasmids and transfections

The Orai1 mutants employed for electrophysiology were engineered into a pEYFP-N1 vector (Clontech) to produce C-terminally tagged Orai1-YFP proteins (*Navarro-Borelly et al., 2008*). mCherry-STIM1 and CFP-CAD were kind gifts of Dr. R. Lewis (Stanford University, USA). All mutants were generated by the QuikChange Mutagenesis Kit (Agilent Technologies) and the mutations were confirmed by DNA sequencing. For electrophysiology, the indicated Orai1 constructs were transfected into HEK293-H cells either alone (200 ng DNA per coverslip) or together with STIM1 (100 ng Orai1 and 500 ng STIM1 DNA per coverslip). For FRET and confocal microscopy experiments, cells were transfected with Orai1-YFP alone (200 ng DNA per coverslip) or with CFP-CAD constructs (100 ng each per coverslip). All transfections were performed using Lipofectamine 2000 (Thermo Fisher Scientific) 24–48 hr prior to electrophysiology or imaging experiments.

## Solutions and chemicals

The standard extracellular Ringer's solution used for electrophysiological experiments contained 130 mM NaCl, 4.5 mM KCl, 20 mM $CaCl_2$, 10 mM tetraethylammonium chloride (TEA-Cl), 10 mM D-glucose, and 5 mM HEPES (pH 7.4 with NaOH). For the FRET and confocal imaging studies, the Ringer's solution contained 2 mM $CaCl_2$ and 150 mM NaCl with the other components as above. The 110 mM $Ca^{2+}$ solution contained 110 mM $CaCl_2$, 10 mM D-glucose, and 5 mM HEPES (pH 7.4 with NaOH). The DVF Ringer's solution contained 150 mM NaCl, 10 mM HEDTA, 1 mM EDTA, 10 mM TEA-Cl and 5 mM HEPES (pH 7.4). The internal solution contained: 135 mM Cs aspartate, 8 mM $MgCl_2$, 8 mM Cs-BAPTA, and 10 mM HEPES (pH 7.2 with CsOH).

## Electrophysiology

Currents were recorded in the standard whole-cell configuration at room temperature on an Axopatch 200B amplifier (Molecular Devices) interfaced to an ITC-18 input/output board (Instrutech).

Routines developed by R. S. Lewis (Stanford) on the Igor Pro software (Wavemetrics) were employed for stimulation, data acquisition and analysis. Data are corrected for the liquid junction potential of the pipette solution relative to Ringer's in the bath (−10 mV). The holding potential was +30 mV. The standard voltage stimulus consisted of a 100 ms step to –100 mV followed by a 100 ms ramp from –100 to +100 mV applied at 1 s intervals. $I_{CRAC}$ was typically activated by passive depletion of ER $Ca^{2+}$ stores by intracellular dialysis of 8 mM BAPTA. All currents were acquired at 5 kHz and low pass filtered with a 1 kHz Bessel filter built into the amplifier. All data were corrected for leak currents collected in 100–200 µM $LaCl_3$.

## Data analysis

Analysis of current amplitudes was typically performed by measuring the peak currents during the −100 mV pulse. Specific mutants were categorized as gain-of-function if their currents exceeded 2 pA/pF, which is more than ten times the current density of WT Orai1 without STIM1. Reversal potentials were measured from the average of several leak-subtracted sweeps in each cell. Fractional inhibition of current was quantified as: Inhibition=$(1-I_b/I_{Ctrl})$, where $I_b$ is the Orai1 current in the presence of $Cd^{2+}$, and $I_{Ctrl}$ is the Orai1 current prior to application of the blocker ($Cd^{2+}$).

## FRET microscopy

HEK293-H cells transfected with Orai1-YFP and CFP-CAD DNA constructs were imaged using wide-field epifluorescence microscopy on an IX71 inverted microscope (Olympus, Center Valley, PA). Cells were imaged with a 60X oil immersion objective (UPlanApo NA 1.40), a 175 W Xenon arc lamp (Sutter, Novatao, CA), and excitation and emission filter wheels (Sutter, Novato, CA). At each time point, three sets of images (CFP, YFP, and FRET) were captured on a cooled EM-CCD camera (Hamamatsu, Bridgewater, NJ) using optical filters specific for the three images as previously described. Image acquisition and analysis was performed with SlideBook software (Imaging Innovations Inc, Denver, CO). Images were captured at exposures of 100–500 ms with 1 × 1 binning. Lamp output was attenuated to 25% by a 0.6 ND filter in the light path to minimize photobleaching. All experiments were performed at room temperature.

FRET analysis was performed as previously described (*Navarro-Borelly et al., 2008*). The microscope-specific bleed-through constants (a = 0.12; b = 0.008; c = 0.002 and d = 0.33) were determined from cells expressing cytosolic CFP or YFP alone. The apparent FRET efficiency was calculated from background-subtracted images using the formalism (*Zal and Gascoigne, 2004*):

$$E_{FRET} = \frac{F_c}{F_c + GI_{DD}}$$

where $F_c = I_{DA}\, aIAA - dI_{DD}$.

$I_{DD}$, $I_{AA}$, and $I_{DA}$ refer to the background subtracted CFP, YFP, and FRET images, respectively. The instrument dependent $G$ factor had the value 1.85 ± 0.1. E-FRET analysis was restricted to cells with YFP/CFP ratios in the range of 2–6 to ensure that E-FRET was compared across identical acceptor to donor ratios, and measurements were restricted to regions of interest drawn at the plasma membrane.

## Confocal microscopy

HEK293-H cells expressing various Orai1-YFP mutants and CFP-CAD were imaged on an Andor XDI Revolution spinning-disk confocal microscope equipped with a 100X oil immersion objective. Cells were maintained at 37˚C with 5% $CO_2$. Fluorophores were excited with 445 nm (CFP) and 515 nm (YFP) laser diodes with the intensity of laser light attenuated to 15–40% for CFP and 5–30% for YFP. Images were obtained at 512 × 512 pixels at an exposure of 200–500 ms per frame and a slice thickness of 0.8 µm. An average of four frames was used for each image. Images analysis was performed using NIH ImageJ software (NIH, Bethesda, MD).

## Molecular dynamics simulations

Molecular models were constructed using the crystal structure of the *Drosophila melanogaster* Orai1 channel (4HKR) (*Hou et al., 2012*). Missing residues of the M1-M2 loop (amino acids 181 to 190) and the M2-M3 loop (amino acids 220 to 235) were modeled de novo using MODELLER (*Fiser and Sali,*

2003). System preparation was performed using CHARMM-GUI membrane builder (*Jo et al., 2007*). The C terminus was truncated at residue 329 for all chains and the N and C terminus were acetylated and amidated, respectively. The protein was embedded within a hydrated 1-palmitoyl,2-oleoyl-sn-glycero-3-phosphocholine (POPC) bilayer with 150 mM NaCl to obtain a hexagonal cell with box vectors 104.2 × 104.2×126.5 Å. Pore waters were not modeled. The simulation cell consisted of ~112K atoms. Single point mutations were made using CHARMM-GUI to create the H206Y, H206Q, H206C, M173L, and M173F systems. The CHARMM36 force field was used for protein (*Best et al., 2012*; *MacKerell et al., 1998*), ions, and lipids (*Klauda et al., 2010*) along with the TIP3P water model (*Jorgensen et al., 1983*).

All simulations were performed using GROMACS 2016.3 (*Murtola et al., 2015*) without modification to the CHARMM-GUI output parameters (with the exception of additional equilibration steps and extended production simulation length). Lennard-Jones interactions were cut off at 1.2 nm and a force-based switching function with a range of 1.0 nm was used. Electrostatic interactions were calculated using particle-mesh Ewald (*Darden et al., 1993*; *Essmann et al., 1995*) with a real-space cut-off distance of 1.2 nm. Nonbonded interactions were calculated using Verlet neighbor lists (*Páll and Hess, 2013*; *Verlet, 1967*). All simulations were performed at constant temperature (323.15 K) and pressure (one atm) using the Nosé-Hoover thermostat (*Hoover, 1985*; *Nosé, 1984*) with temperature coupling of 1.0 ps and the Parrinello-Rahman barostat (*Nosé and Klein, 1983*; *Parrinello and Rahman, 1980*) with a time constant of 5.0 ps, respectively. All hydrogen bonds were constrained using the LINCS algorithm (*Hess, 2008*). The integration time step was two fs. All systems followed the standard energy minimization and six-step equilibration procedure of CHARMM-GUI (*Jo et al., 2007*), followed by two successive 10 ns protein-restrained simulations conducted in the NPT ensemble. In these equilibration steps, position restraints were applied to main chain backbone atoms and then C$\alpha$ atoms, with restraint strength of 1000 kJ mol$^{-1}$ nm$^{-2}$. Twenty simulation repeats were created for WT, H206Q/C, ten simulation repeats were created for H206Y, and thirty repeats were created for M173L/F systems with randomized initial velocities for production simulations. Production simulations were conducted for 400–500 ns for all simulation repeats for an aggregate total of 73.6 μs.

Prior to analysis, all simulation frames were aligned such that the principal axis formed by TM1 helix C$_\alpha$ atoms were aligned to the box vector *z*. Analysis was performed on all simulation frames spaced at 1.0 ns after removing the first 100 ns of data from each simulation repeat. All axial coordinates were measured with respect to the center of mass of the pore helix C$_\alpha$ atoms (residues 141 to 174). Axial histograms of water oxygen atoms, Na$^+$, and Cl$^-$, was computed within a cylinder of radius of 10 Å centered at the pore center of mass. Error bars were computed using the standard error of mean over all simulation repeats. Pearson correlation coefficients were computed using observables measured at all data points for all simulation repeats, with all subunit specific properties averaged across all six subunits. Potential mean force plots in *Figure 7B* were generated from the M173-F171 distance distributions, where the energy change is denoted as *W(d)–W(0)* and the equilibrium constant is represented by *p(d)/p(d₀)*:

$$W(d) - W(0) = -RTln\left(\frac{p(d)}{p(d_0)}\right)$$

where *p(d)* is the probability distribution of the distance d between M173 and F171, *d₀* is an arbitrary reference separation, *R* is the universal gas constant, and *W(d)* is the potential of mean force (PMF) or free energy at *d*.

## Acknowledgements

We thank members of the Prakriya laboratory for helpful discussions. Molecular simulations conducted in this work were enabled by supercomputing resources and support provided by SciNet and Compute Canada (www.computecanada.ca). This research was supported by National Institutes of Health grants NS057499 and GM114210 (to M Prakriya), Canadian Institutes of Health Research grant MOP130461 (to R Pomès), and National Institutes of Health Predoctoral Fellowship F31NS101830 (to PS-W Yeung). Imaging was performed at the Northwestern University Center for Advanced Microscopy generously supported by NCRR 1S10 RR031680. The authors declare no competing financial interests.

## Additional information

### Funding

| Funder | Grant reference number | Author |
|---|---|---|
| National Institutes of Health | NS057499 | Murali Prakriya |
| National Institutes of Health | F31NS101830 | Priscilla S-W Yeung |
| Canadian Institutes of Health Research | MOP130461 | Régis Pomès |
| National Institutes of Health | GM114210 | Murali Prakriya |

The funders had no role in study design, data collection and interpretation, or the decision to submit the work for publication.

### Author contributions

Priscilla S-W Yeung, Conceptualization, Data curation, Formal analysis, Funding acquisition, Validation, Investigation, Methodology, Writing - original draft, Writing - review and editing; Christopher E Ing, Data curation, Software, Formal analysis, Investigation, Methodology, Writing - original draft, Writing - review and editing; Megumi Yamashita, Data curation, Formal analysis, Investigation, Writing - review and editing; Régis Pomès, Conceptualization, Resources, Supervision, Funding acquisition, Investigation, Methodology, Writing - original draft, Writing - review and editing; Murali Prakriya, Conceptualization, Resources, Formal analysis, Supervision, Funding acquisition, Investigation, Methodology, Writing - original draft, Project administration, Writing - review and editing

### Author ORCIDs

Priscilla S-W Yeung (iD) https://orcid.org/0000-0001-5400-8639
Christopher E Ing (iD) https://orcid.org/0000-0001-6947-5731
Régis Pomès (iD) https://orcid.org/0000-0003-3068-9833
Murali Prakriya (iD) https://orcid.org/0000-0003-0781-4480

### Decision letter and Author response

Decision letter https://doi.org/10.7554/eLife.60751.sa1
Author response https://doi.org/10.7554/eLife.60751.sa2

## Additional files

### Supplementary files

• Transparent reporting form

### Data availability

All data generated or analysed during this study are included in the manuscript and supporting files.

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
