## [Decision Letter]

**Acceptance summary:**

The physical mechanisms by which ion channels open and close to gate ion flow across membranes are fundamental to many physiological processes. The authors showed previously that Orai calcium channels open by a twisting motion that rotates hydrophobic amino acid side chains (phenylalanines) out of the pore. Here, they reveal a previously unknown step in which the rotated phenylalanines engage sulfur-containing methionines, and that this interaction is required to keep the channel open. This work significantly extends our understanding of how calcium signaling via this important class of channels is regulated, and more broadly presents the first example of sulfur-aromatic interactions in ion channel gating.

**Decision letter after peer review:**

Thank you for submitting your article "A sulfur-aromatic gate latch is essential for opening of the Orai1 channel pore" for consideration by *eLife*. Your article has been reviewed by three peer reviewers, including Richard S Lewis as the Reviewing Editor and Reviewer #1, and the evaluation has been overseen by Kenton Swartz as the Senior Editor. The following individual involved in review of your submission has agreed to reveal their identity: Michael D Cahalan (Reviewer #3).

The reviewers have discussed the reviews with one another and the Reviewing Editor has drafted this decision to help you prepare a revised submission.

Summary:

The mechanisms that control the activity of store-operated calcium channels are a central focus in the field of cellular calcium signaling. This paper describes a novel motif that is critical for the activation of Orai1 channels by the ER calcium sensor STIM1. In previous work, the authors demonstrated that channel opening results from the rotation of a hydrophobic gate comprising six F99 side chains out of the pore, allowing water to enter and promote calcium conduction. Here, the authors provide strong evidence that a sulfur-aromatic interaction between F99 and M101, termed the "gate latch," promotes opening of the hydrophobic gate. Combining molecular dynamics simulations with the experimental introduction of metal ion bridges, they show that the F99-M101 interaction stabilizes the open state of the channel, and without it the channel cannot open even when bound by STIM1. A notable strength of the paper is the effective complementarity of experimental and computational studies which show that the length, flexibility, and chemistry of M101 is essential for its role in gating. While sulfur-aromatic interactions are known to stabilize proteins, this study is the first example of sulfur-aromatic interactions involved in channel gating. Together with previous work by the authors, this paper creates a dynamic view of motions and atomic interactions that underlie the fundamental transitions between closed and open states of the pore.

Essential revisions:

All three reviewers were enthusiastic about the paper, in particular with regard to the powerful combination of experimental and computational approaches, and mostly have only suggestions for clarifying nomenclature and for further discussion of the results to broaden the scope of the conclusions.

1) It is assumed, but not shown, that as the channel opens M101 rotates along with F99 to enable it to contact F99 on the adjacent helix. This should be extracted from the MD simulations and plotted as in Figure 2—figure supplement 1F.

2) The inhibitory effect of M101L on the constitutively open mutant channel H134S suggests that M101-F99 stabilizes the open state. There are other constitutively active mutants with reduced ion selectivity that do not involve rotation of F99, such as V102C. One prediction of the present study is that the M101-F99 latch would not operate in such a mutant, and hence would be unaffected by the M101L mutation. This experiment is optional, as it is not needed to bolster the main conclusion, but it would serve to distinguish the two types of constitutively active mutants and emphasize that helical rotation is needed to engage the latch.

3) In a previous paper (Yeung et al., 2018), the authors showed that M101C did not activate Orai1 even though it destabilizes the closed state, presumably because of an inability to stabilize the open state. Unlike M101C, F187C was previously shown to activate the channel. Does this imply that F187 normally stabilizes the closed state in ways beyond its interaction with M101C? Please comment.

4) It is interesting that the F99C/M101C channel activated by Cd^2+^ is non-selective (Figure 3), while the same channel activated by STIM1 is Ca^2+^-selective (Figure 4). This would seem to suggest that trapping F99 in a rotated conformation with Cd^2+^ does not rotate the E106 selectivity filter to provide Ca^2+^ selectivity. Please comment.

5) Subsection “Enhancing M101 and F99 interaction boosts F99C/M101C Orai1 channel activity”, third paragraph, Figure 3D. It would be helpful to plot the absolute current levels for the 3 experiments, to provide a sense of how well STIM1 activates WT Orai1 compared to the Cd^2+^-treatment of F99C/M101C. If Cd^2+^ doubles the size of the STIM-activated current, does this suggest that the latch is normally engaged about half the time?

6) Discussion: Some explanatory detail should be given for the calculation of free energy stabilization. It is not obvious where the value of 0.5-1 kcal/mol comes from.

7) The manuscript necessarily uses the *Drosophila* Orai residue numbering in discussing the MD simulations and the human Orai numbering for the electrophysiological data. While the display of both residue numbers in the figures is handled well, the text is sometimes confusing. For example, the manuscript mentions only F99 and M101 in the first few pages, then abruptly switches to *Drosophila* numbering in the Introduction, without flagging the change for readers. (The human Orai1 numbering follows, in parentheses, but the transition could be accomplished more deftly.) The subheading “Molecular dynamics simulations demonstrate F171-M173 sulfur-aromatic interactions involving the hydrophobic gate in activated states” refers to F171 and the next two lines of text refer to F99, without reminding readers that the two phenylalanine residues occupy the same position in the channel. Minimal rewriting could smooth these transitions.

8) In the first paragraph of the Results, the authors use a mutation in TM2 while focusing on residues in TM1 and TM3. The authors should orient the reader as to where H134 is located somewhere in the subsection “Molecular dynamics simulations demonstrate F171-M173 sulfur-aromatic interactions involving the hydrophobic gate in activated states”, and whether it is well understood how it causes channel opening.

9) Figure 3C nicely shows I-V plots with reversal potentials near zero, in the Cd^2+^-potentiated current after removing Cd^2+^, implying a lack of Ca^2+^ selectivity in the double mutant F99C/M101C after Cd^2+^ is added. What does the I-V look like just before removing Cd^2+^? It may be confusing to call this a CRAC current when it seems to be non-selective. A very interesting result, particularly when combined with partial occlusion by STIM1 gating. It might be nice to show the I-V curve of the same mutant in 110 Ca^2+^ external solution (panel F).

10) The GOF channels with smaller residues at position 187 produced nice CRAC-like I-V curves. What does this tell us?

11) The videos strikingly demonstrate the residues interacting. In videos comparing the two open-state channels, the M173F (Figure 6—video 2) constitutively open pore looked significantly smaller (F171 points toward the center of the pore to a greater degree) than in the H206C constitutively open pore (Figure 2—video 2). Are there any functional differences in ion selectivity?

---

## [Author Response]

Essential revisions:All three reviewers were enthusiastic about the paper, in particular with regard to the powerful combination of experimental and computational approaches, and mostly have only suggestions for clarifying nomenclature and for further discussion of the results to broaden the scope of the conclusions.1) It is assumed, but not shown, that as the channel opens M101 rotates along with F99 to enable it to contact F99 on the adjacent helix. This should be extracted from the MD simulations and plotted as in Figure 2—figure supplement 1F.

Thank you for this interesting suggestion. We analyzed the rotation of M101 (M173 in dOrai) using the C_⍺_ atom of M101 and the results indicate that M101 rotation increases with channel activity in a very similar fashion as the rotation of F99. The order of increasing M101 C_⍺_ rotation is H206Y<WT<H206Q<H206C in dOrai simulations, which is identical to the order of channel activity and pore hydration of these mutants (Figure 2—figure supplements 1 and 2). Importantly, a scatterplot of M101 vs. F99 angle in Figure 2—figure supplement 2F which shows that the two are tightly correlated (Pearson coefficient 0.88) confirming that as the channel opens, M101 rotates along with F99 to enable contact formation between the two residues from adjacent helices.

2) The inhibitory effect of M101L on the constitutively open mutant channel H134S suggests that M101-F99 stabilizes the open state. There are other constitutively active mutants with reduced ion selectivity that do not involve rotation of F99, such as V102C. One prediction of the present study is that the M101-F99 latch would not operate in such a mutant, and hence would be unaffected by the M101L mutation. This experiment is optional, as it is not needed to bolster the main conclusion, but it would serve to distinguish the two types of constitutively active mutants and emphasize that helical rotation is needed to engage the latch.

Thank you for suggesting this interesting experiment. To test this idea, we generated the V102CM101L double mutant and compared currents in this mutant to the V102C single mutant (Figure 5—figure supplement 2B). Surprisingly, we found that the addition of M101L significantly attenuated the activity of the V102C leaky pore mutant, implying that the gate latch is still involved in the opening of V102C channels. Unlike H134S, the constitutive activity of V102C is primarily driven by an increase in pore hydration secondary to lowering of the energetic barrier presented by the hydrophobic gate. However, because pore hydration and pore helix rotation are tightly correlated (Yamashita et al., 2017; Yeung et al., 2018), the elimination of the F99-M101 interaction in M101L channels may inhibit pore helix rotation and thus abrogate channel activity. We have added this result to Figure 5—figure supplement 2B.

3) In a previous paper (Yeung et al., 2018), the authors showed that M101C did not activate Orai1 even though it destabilizes the closed state, presumably because of an inability to stabilize the open state. Unlike M101C, F187C was previously shown to activate the channel. Does this imply that F187 normally stabilizes the closed state in ways beyond its interaction with M101C? Please comment.

Yes, thank you for pointing out this subtle but potentially interesting point. The F187-M101 interaction, which we suggest is essential for stabilizing the closed channel state, can be disrupted either by mutating M101 or mutating F187. As you point out, the M101C mutation did not activate Orai1 even though it would be expected to disrupt the closed state F187-M101 interaction because M101 is also important for promoting channel activation as a part of the latch. Yet, the F187C mutation overcomes this effect and can open the double mutant. Clearly, this suggests that the F187 “brake” residue mediates its braking effect not only through interactions with M101, but potentially through an additional interaction with a second partner. The most likely candidate for this second braking effect may be through a potential interaction with L96, which is part of the hydrophobic clamp located one turn below M101 on the non-pore facing side of TM1. We have added a sentence to indicate such a second role in the text (subsection “M101 is essential for STIM1-mediated Orai1 channel activation”, last paragraph).

4) It is interesting that the F99C/M101C channel activated by Cd^2+^ is non-selective (Figure 3), while the same channel activated by STIM1 is Ca^2+^-selective (Figure 4). This would seem to suggest that trapping F99 in a rotated conformation with Cd^2+^ does not rotate the E106 selectivity filter to provide Ca^2+^ selectivity. Please comment.

Actually, the F99C/M101C channel is non-selective without STIM1 and *remains* non-selective in the presence of STIM1. The current-voltage relationships shown in Figure 4 are of the Ca^2+^ selective M101C/F187C mutant. We have added labels to the panels to prevent confusion and improve the clarity of this point.

5) Subsection “Enhancing M101 and F99 interaction boosts F99C/M101C Orai1 channel activity”, third paragraph, Figure 3D. It would be helpful to plot the absolute current levels for the 3 experiments, to provide a sense of how well STIM1 activates WT Orai1 compared to the Cd^2+^-treatment of F99C/M101C. If Cd^2+^ doubles the size of the STIM-activated current, does this suggest that the latch is normally engaged about half the time?

On the surface, yes, this seems like a reasonable conclusion and indeed, Cd^2+^ does enhance the current two-fold in STIM1-activated F99C/M101C channels (~1.5 pA/pF in the absence and 3.2 pA/pF in the presence of Cd^2+^). This could be interpreted (with some assumptions) that in the absence of Cd^2+^, the latch is engaged about 50% of the time as in the presence of Cd^2+^. We are hesitant to make too much of this result however, because the F99C/M101C mutant is not strongly activated by STIM1, with current levels of only about 3-4 pA/pF at steady-state compared to its constitutive activation level of 1.5 pA/pF at whole-cell break-in. This is despite the fact that STIM1 binds normally to F99C/M101C channels as assessed by FRET. We have seen in many previous experiments that the single M101C mutant has a LOF phenotype (as expected from partial loss of the latch function), and F99C channels also have a partial LOF phenotype (again likely to due to inability to engage with the M101 latch) (F99C current amplitudes of -6 pA/pF compared to -35 pA/pF in WT channels; McNally et al., 2009). Therefore, it is no surprise that the double mutant is not effectively activated by STIM1.

That said, STIM1 clearly blunts the extent of Cd^2+^-mediated potentiation of F99C/M101C channels (Figure 3D). This clearly indicates that the conformation change imposed by STIM1 limits Cd^2+^’s ability to further potentiate F99C/M101C activation and we have reasonably interpreted that this conformational change is the pore helix rotation which puts F99C and M101C facing each other. In this orientation, formation of the metal ion bridge between the two positions does little to further enhance the current. We have now added this potential interpretation together with the caveat noted above in the relevant Results section (subsection “Enhancing M101 and F99 interaction boosts F99C/M101C Orai1 channel activity”, third paragraph).

6) Discussion: Some explanatory detail should be given for the calculation of free energy stabilization. It is not obvious where the value of 0.5-1 kcal/mol comes from.

Agreed, and thanks for the suggestion! We have now added a gating transition scheme in Figure 7C and a much more expansive discussion of these calculations (Discussion). In short, we use 7Å as the cutoff to transform the continuous distance probability curves in Figure 2E into binary contact versus non-contact states. We then take the ratios of cumulative probability of the distances in the different variants <7 Å and >7 Å as proxies for the relative occupancies of the contact and noncontact states in the closed (WT and H206Y) and open (H206Q and H206C) channels. Finally, we convert these relative occupancy ratios into energy values using a modified Boltzmann’s equation to estimate an overall energy of ~0.8 kcal/mol.

7) The manuscript necessarily uses the *Drosophila* Orai residue numbering in discussing the MD simulations and the human Orai numbering for the electrophysiological data. While the display of both residue numbers in the figures is handled well, the text is sometimes confusing. For example, the manuscript mentions only F99 and M101 in the first few pages, then abruptly switches to *Drosophila* numbering in the Introduction, without flagging the change for readers. (The human Orai1 numbering follows, in parentheses, but the transition could be accomplished more deftly.) The subheading “Molecular dynamics simulations demonstrate F171-M173 sulfur-aromatic interactions involving the hydrophobic gate in activated states” refers to F171 and the next two lines of text refer to F99, without reminding readers that the two phenylalanine residues occupy the same position in the channel. Minimal rewriting could smooth these transitions.

Thank you for this comment. We have removed the dOrai residue numbers from the subheading and added some transitions to help alert readers to the differences in residue numbering.

8) In the first paragraph of the Results, the authors use a mutation in TM2 while focusing on residues in TM1 and TM3. The authors should orient the reader as to where H134 is located somewhere in the subsection “Molecular dynamics simulations demonstrate F171-M173 sulfur-aromatic interactions involving the hydrophobic gate in activated states”, and whether it is well understood how it causes channel opening.

Thank you for this suggestion. We have added a sentence explaining the location of H134 and its postulated role in channel gating in the Results (subsection “Molecular dynamics simulations reveal sulfur-aromatic interactions involving the hydrophobic gate in activated states”) and we hope will help orient the reader.

9) Figure 3C nicely shows I-V plots with reversal potentials near zero, in the Cd^2+^-potentiated current after removing Cd^2+^, implying a lack of Ca^2+^ selectivity in the double mutant F99C/M101C after Cd^2+^ is added. What does the I-V look like just before removing Cd^2+^? It may be confusing to call this a CRAC current when it seems to be non-selective. A very interesting result, particularly when combined with partial occlusion by STIM1 gating. It might be nice to show the I-V curve of the same mutant in 110 Ca^2+^ external solution (panel F).

Thank you for these suggestions. We have added IV curves of Cd^2+^-gated current of F99C/M101C currents both during the Cd^2+^ application and after washout of Cd^2+^ (Figure 3C) in 20 mM Ca^2+^ and in 110 Ca^2+^ external solution (Figure 3F). The I-Vs show that the current is non-selective reversing close to 0 mV. Also, your point about calling it CRAC current is noted, we now call it the mutant Orai1 current (subsection “Enhancing M101 and F99 interaction boosts F99C/M101C Orai1 channel activity”).

10) The GOF channels with smaller residues at position 187 produced nice CRAC-like I-V curves. What does this tell us?

The data suggest that the gain-of-function F187 mutations put the channel pore likely in a similar conformation as the wildtype channels gated by STIM1, analogous to the pore conformations of H134S channels that we characterized previously (Yeung et al., 2018). However, we should note that the F187X GOF mutants produce significantly smaller currents than the H134S/C variants, suggesting a lower channel Po. We think this is in line with our hypothesis that without the F187 brake, the channel has a tendency to switch into a Ca^2+^-selective open state.

11) The videos strikingly demonstrate the residues interacting. In videos comparing the two open-state channels, the M173F (Figure 6—video 2) constitutively open pore looked significantly smaller (F171 points toward the center of the pore to a greater degree) than in the H206C constitutively open pore (Figure 2—video 2). Are there any functional differences in ion selectivity?

There was no statistically significant difference in the reversal potentials of M101F (44.9 ± 5.7 mV) and H134C (42.0 ± 1.9 mV). However, H134C has a much larger current amplitude (-24.4 ± 3.0 pA/pF) than M101F (-7.7 ± 1.9 pA/pF), consistent with the larger pores observed in the MD simulations of dOrai H206C channels. We have also added two panels in Figure 6 showing the distances between M173L and M173F C_b_ with the aromatic rings of F171 and F259. We show quantitatively that M173L has decreased interaction with F171, whereas the M173F mutant displays increased interactions with both F171 and F259 by these metrics.